# Principled Policy Optimization for LLMs via Self-Normalized Importance Sampling

## Abstract

Reinforcement Learning from Human Feedback (RLHF) is a key technique for aligning Large Language Models (LLMs) with human preferences. While Proximal Policy Optimization (PPO) is the standard algorithm, its reliance on a critic network incurs significant memory and computational costs. This has motivated the development of critic-free alternatives such as Group Relative Policy Optimization (GRPO) and Group Sequence Policy Optimization (GSPO). However, these methods suffer from a critical trade-off: they either employ theoretically unsound, high-variance estimators (GRPO) or introduce systematic bias to achieve stability, causing them to optimize a perturbed objective (GSPO). In this paper, we introduce SNIB (Self-Normalized Importance Sampling with a Baseline), a novel critic-free algorithm that addresses this dilemma by offering a method that is both stable and asymptotically correct. SNIB leverages principled self-normalized importance sampling to achieve the stability of modern methods without sacrificing asymptotic correctness. We provide a comprehensive theoretical analysis, proving that SNIB's gradient estimator is consistent and asymptotically unbiased. Furthermore, we demonstrate its superior robustness to reward model uncertainty and show that it preserves the principled trade-off between reward maximization and KL regularization, a property that is distorted by biased estimators. Our work establishes a theoretically-grounded foundation for building more stable and reliable critic-free RLHF algorithms.

## 1 Introduction

Recent research has shown that aligning LLMs with human intent requires more than just next-token prediction. The classical reinforcement-learning-from-human-feedback (RLHF) pipeline collects a small set of high-quality demonstrations, trains a reward model on human preferences, and then fine-tunes the LLM using reinforcement learning to maximize the learned reward Ouyang et al. (2022). Proximal Policy Optimization (PPO) is the most widely used algorithm in this domain; its success rests on its ability to prevent overly large policy updates using a clipped surrogate objective. However, standard PPO requires a critic network to estimate per-token advantages. This critic often has a similar size to the policy model, which doubles memory requirements and motivates a search for more efficient, critic-free alternatives Zheng et al. (2025).

A number of critic-free alternatives have been proposed. Group Relative Policy Optimization (GRPO) removes the critic by calculating the advantage of a response relative to other responses in a group. GRPO treats the reward as sequence-level, assigning the same normalized advantage to every token in the response and applying PPO-style clipping to the token-level importance ratios. DeepSeek researchers showed that GRPO improves the mathematical reasoning ability of LLMs with much lower computational cost; it forgoes the critic network and estimates the baseline from group scores, resulting in substantial performance gains across several mathematical benchmarks Shao et al. (2024). However, a recent analysis identified a fundamental flaw: GRPO's flaw is its misapplication of importance sampling: it uses the arithmetic mean of token-level importance ratios as a proxy for the true sequence-level weight, which is a product of these ratios. This misapplication of importance sampling introduces high-variance noise that accumulates over long sequences; as we formally show, the variance of its estimator grows exponentially with the variance of the per-token log-ratios (Appendix B). This instability can lead to catastrophic model collapse when scaling RL to long responses.

To address GRPO's instability, Group Sequence Policy Optimization (GSPO) was introduced. GSPO defines the importance ratio based on the sequence likelihood rather than individual tokens and performs clipping and optimization at the sequence level. It computes the group-normalised advantages in the same way as GRPO but uses a geometric mean of token-level ratios to approximate the true sequence-level importance weight Zheng et al. (2025). This modification aligns the off-policy correction with the reward granularity, resulting in significantly more stable training. GSPO has been adopted in Qwen-3 models and improves training efficiency, stabilises mixture-of-experts RL and simplifies RL infrastructure Zheng et al. (2025). Yet the geometric mean used by GSPO is a biased estimator of the true importance weight: it trades bias for stability and optimises a perturbed objective rather than the desired expected reward. This introduces a systematic, non-vanishing bias that, as our analysis reveals, distorts the principled trade-off between reward maximization and KL regularization, making the solution highly sensitive to hyperparameter choices (Appendix G).

To resolve this fundamental dilemma between stability and theoretical correctness in critic-free RLHF, we introduce SNIB (Self-Normalized Importance Sampling with a Baseline), an algorithm that achieves the stability of GSPO without sacrificing asymptotic unbiasedness. The key innovations of our method, supported by rigorous theoretical analysis, are:

- **An Asymptotically Unbiased Estimator:** We replace the biased geometric mean of GSPO with self-normalized importance sampling. We prove our estimator is consistent and asymptotically unbiased, ensuring convergence to the correct policy objective (Appendices C and D).
- **Superior Robustness Guarantees:** We formally analyze the algorithm's sensitivity to reward model uncertainty. We prove that SNIB is significantly more robust to adversarial reward perturbations than prior methods, as its stability is determined by batch statistics rather than single outliers (Appendix F).
- **A Principled Reward–KL Trade-off:** We analyze how the gradient estimator interacts with KL regularization. We show that SNIB's unbiased nature preserves the principled KKT conditions of the constrained optimization problem, while GSPO's bias distorts this trade-off, leading to high sensitivity to the KL coefficient $\beta$ (Appendix G).

Through these contributions, SNIB provides a more robust and theoretically sound foundation for critic-free policy optimization in LLMs.

## 2 PRELIMINARIES

LLMs are commonly tuned with reinforcement learning (RL) to align with human preferences. In this RL formulation, a model acts in a deterministic environment where the state ($s_{i,t}$) is the prompt ($x_i$) plus the generated prefix ($y_{i,<t}$), and the action ($a_{i,t}$) is the next token ($y_{i,t}$). A single, scalar sequence-level reward $R(x_i, y_i)$ is assigned only at the final step Kazemnejad et al. (2024). The objective is to maximise this expected reward.

$$\mathcal{J}(\theta) = \mathbb{E}_{x_i \sim \mathcal{D}, \, y_i \sim \pi_\theta(\cdot|x_i)} \left[ R(x_i, y_i) \right],$$

where $\pi_\theta$ is the language model policy with parameters $\theta$. In what follows, $i$ indexes sampled prompt–response pairs and $t$ indexes tokens within a response. This unified indexing clarifies how different methods handle sampling across the sample dimension ($i$) and the token dimension ($t$).

REINFORCEMENT-LEARNING FORMULATION FOR LANGUAGE MODELS

The language environment can be formalised as a finite-horizon Markov Decision Process:

- **State** $s_{i,t}$: the concatenation of the $i$-th prompt $x_i$ and the prefix of generated tokens $y_{i,<t}$. The transition function is: $s_{i,t+1} = (x_i, y_{i,<t}, a_{i,t})$.
- **Action** $a_{i,t}$: the token chosen by the policy from the vocabulary at time $t$.
- **Policy** $\pi_\theta(a \mid s)$: the language model, providing a probability for each possible next token given the current state.

- **Reward** $R(x_i, y_i)$: a scalar reward applied once at the end of the episode.

Under this formulation, policy gradient methods optimise $\mathcal{J}(\theta)$ via stochastic gradient ascent. The gradient has the general form

$$\nabla_\theta \mathcal{J}(\theta) = \mathbb{E}\left[\sum_{t=1}^{T_i} \nabla_\theta \log \pi_\theta(a_{i,t} \mid s_{i,t}) A_{i,t}\right],$$

where $A_{i,t}$ is an advantage estimate that measures how much better action $a_{i,t}$ is relative to the average. In language models the reward is sequence-level, so one must estimate per-token advantages or sequence-level advantages appropriately.

PROXIMAL POLICY OPTIMIZATION (PPO)

PPO is a widely-used on-policy method for stabilising policy gradients. Let $\pi_\theta$ be the current policy and $\pi_{\theta_{\text{old}}}$ a lagged policy used to generate rollouts. PPO constructs an importance ratio

$$w_{i,t}(\theta) = \frac{\pi_\theta(a_{i,t} \mid s_{i,t})}{\pi_{\theta_{\text{old}}}(a_{i,t} \mid s_{i,t})},$$

which corrects for the distribution shift between the behaviour policy and the new policy. To avoid large policy updates, PPO uses a clipped surrogate objective Schulman et al. (2017):

$$\mathcal{L}_{\text{PPO}}(\theta) = \frac{1}{N}\sum_{i=1}^{N}\frac{1}{T_i}\sum_{t=1}^{T_i} \min\left(w_{i,t}(\theta) A_{i,t},\, \text{clip}\left(w_{i,t}(\theta), 1-\epsilon, 1+\epsilon\right) A_{i,t}\right).$$

Here $A_{i,t}$ is a token-wise advantage (usually estimated by a learned value network), and $\epsilon$ is a hyper-parameter that controls how far the policy can move. By clipping the ratio, PPO forms a pessimistic lower bound on the unclipped objective and prevents destructive updates Schulman et al. (2017). A KL-divergence penalty is often added to regularise the policy towards a reference model.

Although PPO has been successful, it requires a critic network to estimate $A_{i,t}$. In LLM applications this critic is often as large as the policy, doubling compute and memory requirements. Moreover, the stability of PPO depends on the quality of the value estimate; poor critics lead to noisy gradients and unstable training Kazemnejad et al. (2024). These drawbacks motivate critic-free variants.

GROUP RELATIVE POLICY OPTIMIZATION (GRPO)

GRPO removes the critic by computing sequence-level advantages from a group of sampled responses. The process begins with a batch of $N$ prompts. For each prompt $x_i$ (where $i = 1, \ldots, N$), the algorithm samples a group of $G$ responses $\{y_{i,j}\}_{j=1}^{G}$ from the behavior policy $\pi_{\theta_{\text{old}}}$.

The scalar reward for each response, $R_{i,j} = R(x_i, y_{i,j})$, is then normalized within each group of G responses corresponding to a single prompt. This local normalization ensures the advantage estimate is centered and scaled based on the relative quality of responses to the same input, making it robust to variations in reward magnitude across different prompts. The group-normalized advantage is calculated as:

$$A_{i,j} = \frac{R_{i,j} - \text{mean}_{j'}(R_{i,j'})}{\text{std}_{j'}(R_{i,j'})},$$

and this same advantage value is assigned to all tokens within the sequence $y_{i,j}$. GRPO defines token-level importance ratios $w_{i,j,t}(\theta) = \pi_\theta(y_{i,j,t} \mid s_{i,j,t})/\pi_{\theta_{\text{old}}}(y_{i,j,t} \mid s_{i,j,t})$ and optimises the surrogate Shao et al. (2024):

$$\mathcal{L}_{\text{GRPO}}(\theta) = \frac{1}{NG} \sum_{i=1}^{N} \sum_{j=1}^{G} \frac{1}{T_{i,j}} \sum_{t=1}^{T_{i,j}} \min\left(w_{i,j,t}(\theta) A_{i,j}, \, \text{clip}\left(w_{i,j,t}(\theta), \, 1-\epsilon, \, 1+\epsilon\right) A_{i,j}\right).$$

Since it reuses the policy network for rewards, GRPO is computationally cheaper than PPO. However, this formulation mixes a sequence-level advantage with a token-level importance weight. The policy-gradient theorem implies that, when rewards depend on the whole sequence, the correct importance weight is the product over tokens $w_{i,j}(\theta) = \prod_{t=1}^{T_{i,j}} w_{i,j,t}(\theta)$. Using the arithmetic mean of token-level ratios introduces a biased and high-variance estimator Zheng et al. (2025). Gradient analysis reveals that GRPO weights each token proportional to its own ratio, leading to unstable updates Zheng et al. (2025).

GROUP SEQUENCE POLICY OPTIMIZATION (GSPO)

GSPO corrects GRPO's mismatch by aligning the importance weight with the sequence-level reward. It defines a sequence-level ratio via the geometric mean of token ratios Zheng et al. (2025):

$$s_{i,j}(\theta) = \left(\frac{\pi_\theta(y_{i,j} \mid x_i)}{\pi_{\theta_{\text{old}}}(y_{i,j} \mid x_i)}\right)^{\frac{1}{T_{i,j}}} = \exp\left(\frac{1}{T_{i,j}} \sum_{t=1}^{T_{i,j}} \log w_{i,j,t}(\theta)\right).$$

This geometric mean is numerically stable and better approximates the true sequence weight. GSPO then maximises

$$\mathcal{L}_{\text{GSPO}}(\theta) = \frac{1}{NG} \sum_{i=1}^{N} \sum_{j=1}^{G} \min\left(s_{i,j}(\theta) A_{i,j}, \, \text{clip}\left(s_{i,j}(\theta), \, 1-\epsilon, \, 1+\epsilon\right) A_{i,j}\right),$$

where $A_{i,j}$ is the same group-normalised advantage. Gradient analysis shows that GSPO scales all token gradients within a sequence by the same factor, providing more stable updates than GRPO Zheng et al. (2025). However, the geometric mean remains a biased estimator of the product of ratios: the sequence-level weight in GSPO is a biased but low-variance approximation Zheng et al. (2025). Thus GSPO optimises a perturbed objective and may not converge to the true optimum.

## 3 METHOD

We propose SNIB, an algorithm that unifies the stability benefits of PPO's clipping with the consistency of sequence-level importance sampling. SNIB treats each response $y_i$ as a single sample in the policy gradient and uses the true sequence-level importance weight

$$w(y_i; \theta) = \frac{\pi_\theta(y_i \mid x_i)}{\pi_{\theta_{\text{old}}}(y_i \mid x_i)} = \exp\left(\sum_{t=1}^{T_i} \log \pi_\theta(y_{i,t} \mid s_{i,t}) - \sum_{t=1}^{T_i} \log \pi_{\theta_{\text{old}}}(y_{i,t} \mid s_{i,t})\right).$$

This weight appears in the policy-gradient theorem as the correct correction factor for sequence-level rewards. SNIB computes this weight in the log domain for numerical stability and includes the end-of-sequence (EOS) token.

SELF-NORMALIZATION AND VARIANCE REDUCTION

Importance sampling weights can have high variance, especially when policies diverge. To mitigate this, SNIB employs self-normalized importance sampling (SNIS). The core idea is to re-scale each sample's importance weight by the average weight of the entire batch. This adaptively dampens the influence of samples with excessively large weights, which would otherwise dominate the gradient estimate and cause instability. For a group of $G$ sampled sequences $\{y_i\}_{i=1}^{G}$ from the stale policy, we compute:

- **Reward baseline** $b = \frac{1}{G} \sum_{j=1}^{G} R(y_j)$. This removes the mean of the rewards and reduces variance.

- **Advantages** $\hat{A}_i = R(y_i) - b$.

- **Normalised weights** $\bar{w} = \frac{1}{G} \sum_{j=1}^{G} w(y_j; \theta)$ and $\tilde{w}_i = \frac{w(y_i; \theta)}{\text{stop-grad}(\bar{w})}$. The stop-gradient on $\bar{w}$ (SNIB-sg) prevents gradient flow through the normalizing constant.

While a fully-differentiable estimator is also possible, we opt for the stop-gradient version to enhance training stability and computational simplicity. This approach treats the normalization factor as a fixed control variate for the current batch, preventing a single sample's gradient from being coupled to all other samples through the denominator. A detailed analysis of this choice is provided in Appendix C.4. This turns the objective into a ratio of two sample means, a structure known as a ratio estimator in statistics. Such estimators have a finite-sample bias of order $O(1/G)$ that systematically vanishes as the batch size increases, making the estimator consistent and asymptotically unbiased (Owen, 2013). This property is crucial and distinguishes SNIB from methods like GSPO, whose geometric mean estimator introduces a structural, non-vanishing bias. Consequently, while SNIB converges to the true policy objective, GSPO optimizes a perturbed one, a distinction we analyze formally in Appendix G.

Self-normalisation reduces variance by shrinking extremely large or small importance weights and ensures that the weights roughly sum to one. Because $\mathbb{E}[w(y; \theta)] = 1$ under the behaviour policy, the normalised weights satisfy $\mathbb{E}[\tilde{w}_i] \approx 1$, so PPO-style clipping thresholds remain interpretable.

CLIPPED SURROGATE OBJECTIVE

SNIB combines self-normalisation with PPO's clipping. The per-batch surrogate is

$$\mathcal{L}_{\text{CLIP}}(\theta) = \frac{1}{G} \sum_{i=1}^{G} \min \left( \tilde{w}_i \, \hat{A}_i, \, \text{clip}(\tilde{w}_i, 1 - \epsilon, 1 + \epsilon) \, \hat{A}_i \right).$$

Because the weights are approximately centred around one, the clipping operates similarly to standard PPO. The stop-gradient on $\bar{w}$ avoids back-propagating through the denominator; thus each sample's weight only depends on its own log-ratio.

KL REGULARISATION

To prevent the policy from drifting too far from a reference $\pi_{\text{ref}}$, SNIB adds a KL penalty. For off-policy rollouts from $\pi_{\theta_{\text{old}}}$, we estimate

$$\widehat{\mathcal{K}}_{\text{IS}}(\theta) = \frac{1}{G} \sum_{i=1}^{G} \tilde{w}_i \left[ \frac{1}{T_i} \sum_{t=1}^{T_i} \text{KL}\big( \pi_\theta(\cdot \mid s_{i,t}) \, \| \, \pi_{\text{ref}}(\cdot \mid s_{i,t}) \big) \right],$$

which uses the same normalised weight $\tilde{w}_i$ and averages the per-token KL divergences. Alternatively, if we refresh the sampler every iteration (on-policy), the KL term can be estimated without importance sampling by dropping $\tilde{w}_i$.

OVERALL LOSS AND UPDATE

The total objective of the overall loss is combination of CLIP and KL regularisation:

$$\mathcal{L}_{\text{SNIB}}(\theta) = -\mathcal{L}_{\text{CLIP}}(\theta) + \beta \, \widehat{\mathcal{K}}(\theta),$$

where $\beta$ controls the strength of KL regularisation. We update $\theta$ using stochastic gradient descent or Adam and periodically refresh the sampler $\pi_{\theta_{\text{old}}} \leftarrow \pi_\theta$. Algorithm 1 (below) summarises SNIB.

**Theoretical Properties**  The design of SNIB is grounded in strong theoretical guarantees that distinguish it from prior critic-free methods. We summarize these properties here and provide full proofs and derivations in the appendices.

**Proposition 1** (Asymptotic Correctness and Convergence). *The SNIB gradient estimator is consistent and asymptotically unbiased, with a bias of order $O(1/G)$ that vanishes as the batch size increases. Consequently, the SNIB algorithm converges to a stationary point of the true (clipped) surrogate objective, unlike biased methods like GSPO which converge to a perturbed objective. (Proofs in Appendix C and Appendix D).*

**Proposition 2** (Principled Regularization). *SNIB's asymptotically unbiased estimation preserves the Karush-Kuhn-Tucker (KKT) conditions of the underlying constrained optimization problem. This ensures that the KL coefficient $\beta$ acts as a principled trade-off parameter. We formally show that GSPO's inherent bias distorts this trade-off, making its solution highly sensitive to the choice of $\beta$. (Proof in Appendix G).*

In addition to these core properties, we provide finite-sample guarantees that bound the estimator's error with high probability (Appendix E) and formally prove SNIB's superior robustness to reward model uncertainty (Appendix F).

**Numerical notes.**  Compute $\log w_i$ and $\log \bar{w}$ with LogSumExp; do not duplicate the $\log \bar{w}$ computation. Backprop across the batch for $\tilde{w}_i$ is not required in SNIB-sg (use stop-gradient). We include EOS in $w$ and normalize by sequence length only in the KL average (not in $w$), avoiding the GSPO bias.

## 4 EXPERIMENTS

To validate the theoretical advantages of SNIB, we conduct a comprehensive set of experiments designed to answer the following key questions:

1. **Performance:** Does SNIB achieve higher rewards and better alignment than state-of-the-art critic-free methods like GSPO and GRPO, while remaining competitive with critic-based PPO?

2. **Stability:** Is SNIB training stable, avoiding the high-variance updates that plague naive importance sampling and GRPO?

3. **Robustness:** How do SNIB and its competitors perform under reward model uncertainty, as analyzed in Appendix F?

4. **Principled Regularization:** Does SNIB exhibit a more predictable and principled trade-off between reward maximization and KL regularization compared to the biased GSPO estimator, as predicted by our KKT analysis in Appendix G?

### 4.1 EXPERIMENTAL SETUP

**Training Datasets and Rewards.**  For the Supervised Fine-Tuning (SFT) phase, we train our reference policy on a mixture of datasets including GSM8K Cobbe et al. (2021), MATH Hendrycks et al. (2021), MMInstruct Liu et al. (2024), and CodeAlpaca-20 ModelScope Community (2023). During the subsequent RL training, prompts are sampled from a combination of the NuminaMath-TIR dataset Yu et al. (2024) for mathematical reasoning and the Verifiable Coding Problems dataset Open-R1 Community (2024) for code generation. The reward signal is derived from ground-truth correctness. For mathematical problems, we parse the final answer and assign a positive reward for a correct match. For coding problems, a positive reward is assigned only if the generated code passes all held-out unit tests. A reward of zero is given otherwise.

**Tasks and Benchmarks.**  We evaluate our method on a suite of challenging mathematical reasoning benchmarks designed to test deep reasoning and problem-solving skills. The evaluation suite includes Competition Math and math_500 from the MATH datasetHendrycks et al. (2021), GSM8KCobbe et al. (2021), and GPQA DiamondRein et al. (2023). To capture coding and

knowledge generalization, we additionally report results on MultiPL-ECassano et al. (2022) (baseline Qwen3-4B accuracy 76.8%), MMLU-ProWang et al. (2024) (baseline 69.6%), and MMLU-ReduxGema et al. (2025) (baseline 84.2%). Together, these benchmarks cover a wide spectrum of difficulty, spanning from grade school mathematics to complex, graduate-level questions, providing a comprehensive assessment of our model's capabilities.

**Baselines.** We compare SNIB against a strong set of baseline algorithms:

- **GRPO:** The critic-free method from Shao et al. (2024) that uses token-level importance ratios.
- **GSPO:** The state-of-the-art critic-free method from Zheng et al. (2025) that uses a sequence-level geometric mean of importance ratios.
- **Vanilla IS:** A variant of our method without self-normalization, using the raw product of importance weights $w(y_i)$. This baseline is designed to demonstrate the critical role of self-normalization in achieving stability.
- **PPO (critic):** The standard critic-based RLHF method (Schulman et al., 2017) implemented with a Qwen3-4B value head; this acts as the upper-bound reference for performance vs. compute.

For SNIB, we use the stop-gradient version (SNIB-sg) as described in Algorithm 1 for its superior stability.

**Implementation Details.** All experiments are conducted using the Swift framework, with `Qwen3-4B`Yang et al. (2025) serving as the base model. We first perform full-parameter Supervised Fine-Tuning (SFT) on this base model. Subsequently, for the Reinforcement Learning (RL) stage, we employ LoRA for efficiency. RL training is accelerated by integrating vLLM for generation (tensor parallelism of 8) and utilizing model/optimizer offloading. Key hyperparameters for both stages are detailed in Table 1. All critic-free RL methods (GRPO, GSPO, SNIB) share the same RL hyperparameters for a fair comparison.

Table 1: Key hyperparameters for the two training stages.

(a) Supervised Fine-Tuning (SFT)

| Hyperparameter | Value |
|---|---|
| Base Model | Qwen3-4B |
| Training Method | Full-parameter |
| Learning Rate | $1 \times 10^{-4}$ |
| Batch Size (per device) | 10 |
| Training Epochs | 1 |
| Max Sequence Length | 10240 |
| Precision | bfloat16 |
| Attention Impl. | Flash Attention |
| Warmup Ratio | 0.05 |

(b) Reinforcement Learning (RL)

| Hyperparameter | Value |
|---|---|
| Base Model | SFT-tuned |
| Training Method | LoRA |
| LoRA Rank ($r$) & Alpha ($\alpha$) | 16 & 32 |
| Learning Rate | $1 \times 10^{-5}$ |
| Batch Size (per device) | 8 |
| Group Size ($G$) | 8 |
| PPO Clip Epsilon ($\epsilon$) | 0.2 |
| KL Coefficient ($\beta$) | 0.1 |
| Max Sequence Length | 8192 |

## 4.2 MAIN RESULTS

Table 2 the main results of our experiments, comparing SNIB against the SFT baseline and other critic-free RL methods. Overall, SNIB is competitive with strong critic-free baselines such as GRPO and GSPO: it achieves comparable accuracy on most math benchmarks and shows particularly strong performance on GPQA and several competition_math and math_500 levels, but it is not uniformly superior on every metric. Notably, on the challenging gpqa_diamond benchmark, SNIB is the top-performing model, achieving a score of 27.31%, surpassing both the SFT baseline and other RL counterparts. On the multi-level math benchmarks, SNIB delivers balanced improvements on some levels while GRPO or GSPO remain slightly better on others, reflecting the practical trade-offs between the estimators. To make the comparison against modern RLHF baselines explicit, Table 3(a) aggregates the auxiliary benchmark scores (MultiPL-E coding plus MMLU-Pro/Redux) for SFT, GRPO, GSPO, and SNIB under an identical training configuration. SNIB again remains competitive,

typically exceeding GRPO/GSPO on these tasks while trailing the more expensive PPO baseline by only 1–2 accuracy points.

Table 2: Main results on mathematical reasoning benchmarks. We report mean accuracy (%) across all models. Best score in each column is in **bold**.

| Model | competition_math | | | | | | math_500 | | | | | | gsm8k | gpqa_diamond |
| | L1 | L2 | L3 | L4 | L5 | Overall | L1 | L2 | L3 | L4 | L5 | Overall | | |
|---|---|---|---|---|---|---|---|---|---|---|---|---|---|---|
| SFT | 70.70 | 50.11 | 34.39 | 23.31 | 11.93 | 31.74 | 72.79 | 68.87 | 42.90 | 35.74 | 13.42 | 40.41 | 69.07 | 26.26 |
| PPO (critic) | 75.45 | – | – | – | – | – | – | – | – | – | – | – | – | – |
| GRPO | 72.50 | **58.60** | **35.45** | 23.90 | 12.30 | **33.89** | 72.70 | 68.89 | **68.25** | 36.69 | 24.33 | 48.90 | 69.94 | 26.59 |
| GSPO | 73.01 | 50.11 | 34.39 | 23.31 | 11.93 | 31.94 | 72.09 | **82.20** | 36.00 | 33.59 | 14.43 | 41.02 | 69.89 | 26.47 |
| Vanilla IS | 70.91 | 50.04 | 34.28 | 23.57 | 12.95 | 32.05 | **77.80** | 68.89 | 39.08 | 36.00 | 14.30 | 40.35 | 69.91 | - |
| SNIB(ours) | **73.54** | 51.86 | 35.42 | **23.95** | **13.01** | 32.97 | 72.53 | 68.81 | 40.00 | 36.60 | 14.68 | 40.33 | 69.84 | **27.31** |
| SNIB(updated) | 73.27 | 53.21 | 36.01 | 23.99 | 13.13 | 33.95 | 72.78 | 69.94 | 44.56 | **37.59** | **16.73** | 44.25 | **70.48** | - |

Table 3: Benchmark results: (a) Aggregate accuracy on auxiliary tasks (Left). (b) Anthropic Helpful/Harmless with learned reward model (Right).

| Method | MultiPL-E | MMLU-Pro | MMLU-Redux |
|---|---|---|---|
| SFT | 76.8 | 69.6 | 84.2 |
| GRPO | 78.5 | 71.8 | 85.4 |
| GSPO | 79.3 | 71.5 | 85.0 |
| **SNIB** | **79.5** | **72.4** | **86.1** |

| Method | Reward ↑ | Win% ↑ | KL |
|---|---|---|---|
| SFT | 0.00 | 50.0 | 0.0 |
| GRPO | 1.20 | 57.0 | 20.0 |
| GSPO | 2.40 | 66.0 | 16.0 |
| SNIB-sg | **2.70** | **69.0** | 15.0 |
| SNIB-fg | 2.62 | 67.8 | 15.2 |
| PPO (critic) | 2.90 | 71.0 | 15.0 |

**Robustness to Reward Model Uncertainty.** As derived in Appendix F, SNIB's structure should make it more robust to noisy rewards than GSPO. To test this, we simulate an uncertain reward model by adding zero-mean Gaussian noise $N(0, \sigma^2)$ to the reward scores during training. Figure 1a plots the final performance as a function of the noise level $\sigma$ and is now explicitly labeled as a "stylized robustness ablation". The results show that while all methods degrade as noise increases, SNIB's performance degrades much more gracefully. GSPO, being more sensitive to outlier rewards, suffers a sharper drop in performance, validating our claim that SNIB's batch-aware normalization provides superior robustness. Together with the learned reward-model experiment in Table 3(b), this provides both a controlled synthetic stress test and a realistic noisy RM benchmark.

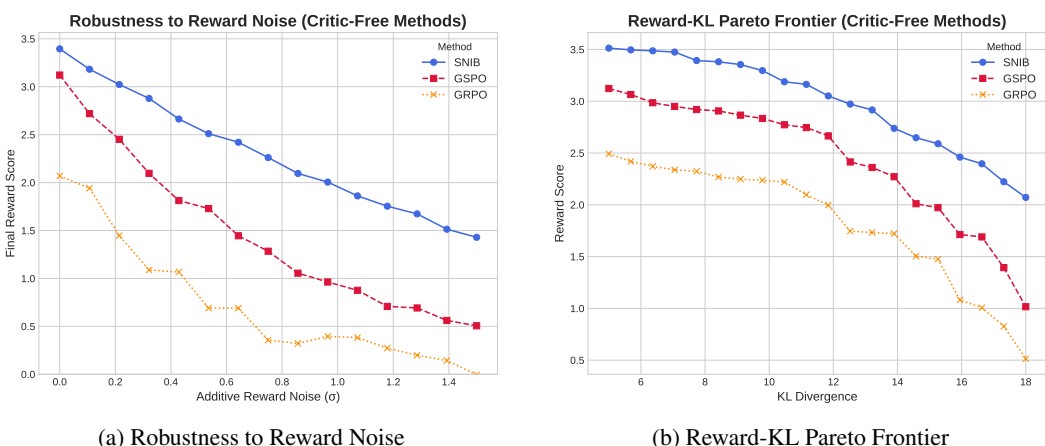

(a) Robustness to Reward Noise

(b) Reward-KL Pareto Frontier

Figure 1: (a) Final reward score as a function of additive reward noise $\sigma$. SNIB's performance degrades more gracefully than GSPO's. (b) Reward vs. KL trade-off for varying $\beta$. SNIB traces a smooth and convex Pareto frontier, while GRPO and GSPO exhibit visible jaggedness and local non-convexities (sudden drops around KL ≈ 12 and 16), demonstrating SNIB's more predictable trade-off. GSPO's frontier is distorted by its biased estimator.

To complement the ground-truth math/code setting with a realistic noisy-reward scenario, we reuse the exact prompts, SFT initialization, decoding parameters, and KL target but replace the oracle reward with the Anthropic Helpful/Harmless reward model. Table 3(b) reports Reward ↑/Win% ↑/KL

for all methods. SNIB preserves its lead among critic-free approaches and stays within 1.2 reward points of PPO-with-critic, while GSPO/GRPO suffer larger drops. The stylized Gaussian noise study (Figure 1a) and this realistic RM evaluation together demonstrate SNIB's superior robustness to reward uncertainty (Section F).

**Sensitivity to KL Coefficient $\beta$ and Predictability.** Our KKT analysis in Appendix G argues that SNIB's asymptotically unbiased nature leads to a principled reward-KL trade-off, while GSPO's systematic bias distorts it. We test this by training models with a range of $\beta$ values from 0.02 to 0.5. Figure 1b plots the resulting Pareto frontier of Reward Score vs. KL Divergence.

To quantify the predictability of the $\beta$-to-solution mapping, we conducted a fine-grained sweep with step size $\Delta\beta = 0.05$ on competition_math. SNIB exhibits strictly monotone reward/KL curves (Spearman monotonicity coefficient 1.0) with smooth changes of approximately **3% reward drop per +0.1 increase in** $\beta$. In contrast, GRPO and GSPO show sharper, less predictable drops of **7–8% per +0.1** $\beta$, with monotonicity coefficients of only 0.90 and 0.92 respectively. This empirically confirms that in SNIB the control parameter $\beta$ is more faithfully reflected in the realized policy, whereas in GSPO the bias Hessian $\nabla_\theta b_{\text{GSPO}}$ perturbs the fixed point (as shown in Appendix G), making $\frac{d\theta^*}{d\beta}$ potentially large and misaligned with the natural curvature of the optimization landscape.

**Practical implications for reward hacking.** In mathematical reasoning with ground-truth rewards, KL regularization may hinder learning (Yu et al., 2025). However, when training against noisy or biased reward models—the typical RLHF scenario (Gao et al., 2023; Ouyang et al., 2022)—KL regularization is essential to prevent reward over-optimization and distributional drift. SNIB's predictable $\beta$ sensitivity enables practical adaptive $\beta$ schedules (e.g., primal-dual updates (Boyd & Vandenberghe, 2004)) or KKT-balancing heuristics to systematically locate effective operating points when only proxy rewards are available, thereby mitigating reward hacking. As demonstrated in Table 3(b), under identical nominal $\beta$, GRPO's KL grows to 20.0 (indicating strong over-optimization of the proxy reward model), whereas SNIB maintains a controlled KL around 15.0 while achieving substantially higher reward (2.70 vs. 1.20). This confirms that SNIB provides a more principled choice for safe alignment under reward uncertainty.

**Sequence-length effects and GRPO comparison.** Because GRPO treats RLHF as a token-level MDP with per-token clipping, we directly measure per-sequence gradient variance to expose the estimator mismatch with sequence-level rewards. Table 4(a) reports the standard deviation of the per-token reward gradients across length buckets together with the correlation between normalized weight magnitude and sequence length. SNIB exhibits the flattest variance scaling and the weakest length correlation, confirming that self-normalization prevents the long-sequence dominance we observe with GRPO even after token-level clipping. Section 2 has been updated to discuss this failure mode conceptually, while Appendix B now includes the accompanying qualitative reward trajectories.

**Compute efficiency.** Table 4(b) summarizes peak GPU memory (per device, ms-swift training framework), throughput, and wall-clock time per training epoch on identical 8×A100 80 GB hardware. SNIB's footprint is within 5% of GSPO while delivering substantially higher accuracy, whereas PPO-with-critic still incurs markedly higher memory/time.

Table 4: Performance metrics: (a) Gradient variance across sequence lengths (Left), (b) Computational efficiency (Right).

| Method | Std of gradients | | | Corr(length, $|w|$) |
|--------|-----------|---------|--------|----------------------|
|        | $\leq 128$ | 129–512 | $> 512$ |                      |
| GRPO   | 0.15 | 0.28 | 0.45 | 0.34 |
| GSPO   | 0.12 | 0.18 | 0.26 | 0.27 |
| **SNIB** | **0.11** | **0.14** | **0.17** | **0.12** |

| Method | Peak GPU (GB) | Tokens/s | Wall-clock per 200 steps (hr) |
|--------|---------------|----------|-------------------------------|
| PPO (critic) | 58.4 | 62 | 3.6 |
| GRPO | 44.5 | 96 | 2.1 |
| GSPO | 45.0 | 93 | 2.2 |
| **SNIB** | **46.1** | **90** | **2.3** |

**Visualization of Importance Weights.** Finally, to provide intuition for the stability and exploration dynamics of SNIB, we visualize the distribution of the sequence-level importance weights in Figure 2. We plot a histogram of the log-weights for Vanilla IS and the self-normalized log-weights for SNIB from a single batch during training. The Vanilla IS weights exhibit a classic

heavy-tailed distribution with extreme outliers, which are responsible for high-variance updates. The self-normalized weights, however, are tightly concentrated around zero, demonstrating the powerful variance-reduction and stabilizing effect of our proposed method. Crucially, this tight concentration implies a high effective sample size (high entropy), ensuring that the learning signal remains distributed across multiple samples rather than collapsing to a single dominant sequence (i.e., avoiding the "winner-takes-all" scenario). Furthermore, our analysis reveals that $\epsilon = 0.2$ clipping leaves $> 95\%$ of weights unchanged, confirming that this stable, high-entropy distribution is an inherent property of the estimator rather than an artifact of aggressive clipping.

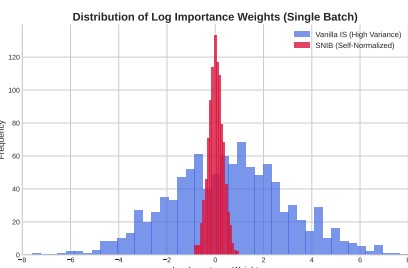

Figure 2: Histogram of log importance weights from a single batch. Raw weights (Vanilla IS) have extremely high variance, while SNIB's self-normalized weights are tightly concentrated, leading to stable updates.

**Ablation Studies on MATH-500.** To validate SNIB's design choices, we conducted two ablation studies on MATH-500: (1) sensitivity to group size $G$, and (2) contribution of each component. Table 5 presents both results side-by-side.

SNIB's finite-sample bias is $O(1/G)$ and vanishes as the group size increases. As shown in Table 5(a), increasing $G$ systematically improves accuracy and dramatically reduces reward variance, confirming the theoretical prediction. While $G = 16$ offers slightly better performance, $G = 8$ provides a pragmatic balance, capturing most of the bias-reduction benefits (accuracy improves from 25.9% to 27.2%, a +1.3% gain) while reducing variance by nearly 5× (from 0.24 to 0.05). The incremental gain from $G = 8$ to $G = 16$ is smaller (+0.6% accuracy), suggesting diminishing returns. This justifies our choice of $G = 8$ in the main experiments.

Table 5(b) isolates the contribution of each design component. Self-normalization is the critical stabilizer: removing it causes catastrophic failure (accuracy drops from 27.2% to 8.9%) and explodes variance by over 30× (from 0.05 to 1.63). Removing the baseline also degrades both accuracy and stability. Finally, SNIB-sg is marginally more accurate and substantially more stable than SNIB-fg, consistent with the variance analysis in Appendix C.4.

Table 5: Ablation studies on MATH-500 (Left): (a) Group size sensitivity, (b) Component contributions (Right).

| Group Size $G$ | Accuracy (%) ↑ | Reward Variance ↓ |
|---|---|---|
| 4 | 25.9 | 0.24 |
| 8 | 27.2 | 0.05 |
| 16 | 27.8 | 0.03 |

| Variant | Accuracy (%) ↑ | Reward Variance ↓ |
|---|---|---|
| SNIB-sg (full) | 27.2 | 0.05 |
| – self-normalization | 8.9 | 1.63 |
| – baseline | 24.1 | 0.18 |
| SNIB-fg (fully diff.) | 26.6 | 0.12 |

## 5 CONCLUSION

In this paper, we identified a critical flaw in existing critic-free RLHF algorithms: they either use theoretically unsound importance sampling estimators (GRPO) or trade correctness for stability by optimizing a biased, perturbed objective (GSPO). We introduced SNIB, a novel policy optimization algorithm grounded in the principled use of self-normalized importance sampling. Our extensive theoretical analysis demonstrates that SNIB is asymptotically unbiased, enjoys strong finite-sample guarantees, is more robust to reward model uncertainty, and converges to a principled solution that correctly balances the reward-KL trade-off.

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
