---

**Algorithm 1** PPO–SNIB (Self-Normalized Importance Sampling with Baseline)

---

**Require:** Initial policy parameters $\theta$; fixed reference policy $\pi_{\text{ref}}$; set sampler $\pi_{\theta_{\text{old}}} \leftarrow \pi_\theta$; batch size $G$; clip $\epsilon$; KL weight $\beta$; optimizer (e.g., Adam) and step size $\alpha$.
1: **for** each training iteration **do**
2:     Sample prompts $\{x_i\}_{i=1}^G \sim \mathcal{D}$.
3:     Generate sequences $\{y_i\}_{i=1}^G \sim \pi_{\theta_{\text{old}}}(\cdot|x_i)$; collect token states $s_{i,t} = (x_i, y_{i,<t})$ (include EOS).
4:     Compute sequence rewards $R_i \leftarrow R(x_i, y_i)$ for all $i$.
5:     Baseline and advantages: $b \leftarrow \frac{1}{G} \sum_{i=1}^G R_i, \quad \hat{A}_i \leftarrow R_i - b.$
6:     Token log-prob sums (current vs sampler):

$$\ell_i^\theta \leftarrow \sum_{t=1}^{T_i} \log \pi_\theta(y_{i,t} \mid s_{i,t}), \quad \ell_i^{\text{old}} \leftarrow \sum_{t=1}^{T_i} \log \pi_{\theta_{\text{old}}}(y_{i,t} \mid s_{i,t})$$

7:     Sequence log-ratio: $u_i \leftarrow \ell_i^\theta - \ell_i^{\text{old}}$.
8:     Log-mean-ratio (numerically stable): $\log \bar{w} \leftarrow \text{LogSumExp}(u_1, \ldots, u_G) - \log G$.
9:     **SNIB-sg normalized weights:** $\tilde{u}_i \leftarrow u_i - \text{sg}[\log \bar{w}], \quad \tilde{w}_i \leftarrow \exp(\tilde{u}_i)$.
10:    Clipped surrogate:

$$\mathcal{L}_{\text{CLIP}}(\theta) \leftarrow \frac{1}{G} \sum_{i=1}^G \min\left( \tilde{w}_i \, \hat{A}_i, \, \text{clip}(\tilde{w}_i, 1 - \epsilon, 1 + \epsilon) \, \hat{A}_i \right)$$

11:    Tokenwise KL to reference on visited states:

$$K_i \leftarrow \frac{1}{T_i} \sum_{t=1}^{T_i} D_{\text{KL}} \left( \pi_\theta(\cdot \mid s_{i,t}) \, \| \, \pi_{\text{ref}}(\cdot \mid s_{i,t}) \right)$$

12:    **Off-policy KL (IS-corrected):** $\mathcal{K}(\theta) \leftarrow \frac{1}{G} \sum_{i=1}^G \tilde{w}_i \, K_i$
       *(If you roll out on-policy, drop $\tilde{w}_i$ and set $\pi_{\theta_{\text{old}}} \leftarrow \pi_\theta$ each iteration.)*
13:    Total loss: $\mathcal{L}(\theta) \leftarrow -\mathcal{L}_{\text{CLIP}}(\theta) + \beta \mathcal{K}(\theta)$.
14:    Update parameters: $\theta \leftarrow \theta - \alpha \nabla_\theta \mathcal{L}(\theta)$.
15:    Periodically refresh sampler: $\pi_{\theta_{\text{old}}} \leftarrow \pi_\theta$.
16: **end for**

---

# A  ALGORITHM

# B  A COMPARATIVE ANALYSIS OF IMPORTANCE SAMPLING ESTIMATORS

This appendix provides a detailed theoretical examination of the importance sampling (IS) estimators that form the basis for modern critic-free RLHF algorithms. We first establish the theoretically correct "gold standard" and then build a clear hierarchy by analyzing the bias and variance of the estimators used in GRPO, GSPO, and our proposed SNIB.

## B.1  THE HIERARCHY OF ESTIMATORS: FROM FLAWED TO PRINCIPLED

**The Gold Standard: True Importance Sampling.**  The objective of policy optimization is to maximize the expected sequence-level reward, $\mathcal{J}(\theta) = \mathbb{E}_{y \sim \pi_\theta}[R(y)]$. In an off-policy setting, the unique, unbiased estimator for this objective relies on the true sequence-level importance weight $w(y)$:

$$w(y) = \frac{\pi_\theta(y|x)}{\pi_{\theta_{\text{old}}}(y|x)} = \prod_{t=1}^T w_t(\theta) \tag{1}$$

Any theoretically sound policy gradient algorithm must correctly utilize this multiplicative weight. However, the product form is numerically unstable for long sequences, as it can easily explode or vanish, making it impractical for direct use. This necessitates approximations, whose properties we analyze below.

**Level 1: GRPO's Arithmetic Mean – Fundamentally Flawed.** GRPO implicitly uses the arithmetic mean (AM) of token-level weights as a proxy: $w_{\text{GRPO}}(y) = \frac{1}{T} \sum_{t=1}^{T} w_t(\theta)$. This choice is theoretically unsound due to both high bias and high variance. As we will show formally in Section A.2, its variance grows exponentially with the variance of the underlying log-ratios, making it highly unstable.

**Level 2: GSPO's Geometric Mean – A Pragmatic but Biased Compromise.** GSPO addresses GRPO's flaws by using the geometric mean (GM), $s(y) = w(y)^{1/T}$, which is computed stably in the log-domain: $s(y) = \exp(\frac{1}{T} \sum_t \log w_t(\theta))$. This significantly reduces variance. However, it introduces a systematic, non-vanishing bias, as the transformation $w(y) \to w(y)^{1/T}$ fundamentally alters the quantity being estimated. The algorithm converges to a stationary point of a perturbed objective, not the true one.

**Level 3: SNIB's Self-Normalization – An Asymptotically Unbiased Solution.** Our proposed estimator, SNIB, achieves stability without sacrificing theoretical consistency by using data-driven self-normalization. For a batch of $G$ samples, the weight for sample $y_i$ is:

$$w_{\text{norm}}(y_i) = \frac{w(y_i)}{\frac{1}{G} \sum_{j=1}^{G} w(y_j)} \tag{2}$$

This approach directly resolves the dilemma. It is a ratio estimator, whose statistical bias is of order $O(1/G)$ and vanishes as the batch size increases. Thus, SNIB is consistent and asymptotically unbiased, ensuring convergence to the correct objective.

### B.2 FORMAL ANALYSIS OF ESTIMATOR VARIANCE

We now formally derive and compare the variance of the GRPO and GSPO estimators. Let the random variables for the log-ratios be $X_t = \log w_t(\theta)$. We assume that for a given sequence, the $X_t$ are independent and identically distributed (i.i.d.) with mean $\mu$ and variance $\sigma^2$. We note that this i.i.d. assumption is a simplification for analytical tractability; in reality, the distribution of $X_t$ is conditioned on the preceding tokens. However, this model effectively captures the fundamental difference in how the estimators aggregate variance, demonstrating the exponential versus linear dependence on the per-token variance that drives their respective instability and stability. The token-level weight is then the random variable $w_t = e^{X_t}$.

First, we derive the mean and variance of $w_t$ using the properties of the log-normal distribution (or the moment-generating function of a normal distribution if we approximate $X_t$ as normal).

$$\mathbb{E}[w_t] = \mathbb{E}[e^{X_t}] = e^{\mu + \sigma^2/2} \tag{3}$$

$$\text{Var}(w_t) = \mathbb{E}[w_t^2] - (\mathbb{E}[w_t])^2 = \mathbb{E}[e^{2X_t}] - (e^{\mu + \sigma^2/2})^2$$

$$= e^{2\mu + (2\sigma)^2/2} - e^{2\mu + \sigma^2} = e^{2\mu + 2\sigma^2} - e^{2\mu + \sigma^2}$$

$$= e^{2\mu + \sigma^2}(e^{\sigma^2} - 1) \tag{4}$$

### B.2.1 VARIANCE OF THE GRPO ESTIMATOR

The GRPO estimator is the sample mean of $w_t$. Since the $w_t$ are i.i.d., the variance of their mean is:

$$\text{Var}(w_{\text{GRPO}}) = \text{Var}\left(\frac{1}{T} \sum_{t=1}^{T} w_t\right) = \frac{1}{T^2} \sum_{t=1}^{T} \text{Var}(w_t) = \frac{1}{T} \text{Var}(w_t) \tag{5}$$

Substituting our expression for $\text{Var}(w_t)$:

$$\text{Var}(w_{\text{GRPO}}) = \frac{1}{T} e^{2\mu + \sigma^2}(e^{\sigma^2} - 1) \tag{6}$$

This result is critical: the variance of the GRPO estimator grows exponentially with the variance of the log-ratios, $\sigma^2$, due to the $(e^{\sigma^2} - 1)$ term. This formally confirms its instability.

### B.2.2 Variance of the GSPO Estimator

The GSPO estimator is $s(y) = e^{\bar{X}}$, where $\bar{X} = \frac{1}{T} \sum_{t=1}^{T} X_t$ is the sample mean of the log-ratios. The mean and variance of $\bar{X}$ are $\mathbb{E}[\bar{X}] = \mu$ and $\text{Var}(\bar{X}) = \sigma^2/T$.

To approximate the variance of the non-linear function $g(\bar{X}) = e^{\bar{X}}$, we use a first-order Taylor expansion, also known as the delta method. The variance of $g(\bar{X})$ is approximated as:

$$\text{Var}(g(\bar{X})) \approx [g'(\mathbb{E}[\bar{X}])]^2 \text{Var}(\bar{X}) \tag{7}$$

In our case, $g'(x) = e^x$, and $\mathbb{E}[\bar{X}] = \mu$. Therefore, $g'(\mu) = e^\mu$. Substituting these into the formula:

$$\text{Var}(s(y)) = \text{Var}(e^{\bar{X}}) \approx (e^\mu)^2 \text{Var}(\bar{X})$$

$$= e^{2\mu} \left( \frac{\sigma^2}{T} \right) \tag{8}$$

So, the approximate variance of the GSPO estimator is:

$$\text{Var}(s(y)) \approx \frac{\sigma^2}{T} e^{2\mu} \tag{9}$$

Comparing Eq. 6 and 9, we see that the variance of the GSPO estimator grows linearly with $\sigma^2$, whereas GRPO's grows exponentially. This formally explains why GSPO is a much more stable, lower-variance estimator. This stability, however, is achieved at the cost of introducing a systematic bias, which SNIB resolves.

## C A Near-Optimal Solution: Self-Normalized IS with a Baseline (SNIB)

We now introduce an advanced estimator that directly tackles the variance of the true importance weight $w(y)$ while retaining theoretical guarantees of consistency. This method combines two powerful statistical techniques: a baseline for reward variance reduction and self-normalization for weight variance reduction.

The objective function for a batch of $G$ samples $\{y_i\}_{i=1}^{G}$ is:

$$\hat{\mathcal{J}}_{\text{SNIB}}(\theta) = \frac{1}{G} \sum_{i=1}^{G} w_{\text{norm}}(y_i) \cdot (R(y_i) - b) \tag{10}$$

where $b$ is a baseline, typically the sample mean of rewards $b = \frac{1}{G} \sum_j R(y_j)$, and $w_{\text{norm}}(y_i)$ is the self-normalized importance weight:

$$w_{\text{norm}}(y_i) = \frac{w(y_i)}{\frac{1}{G} \sum_{j=1}^{G} w(y_j)} \tag{11}$$

The true weight $w(y_i)$ is computed in the log-domain for numerical stability: $w(y_i) = \exp\left( \sum_{t=1}^{T_i} \log w_{i,t}(\theta) \right)$.

### C.1 Bias Analysis: Asymptotic Unbiasedness

The SNIB estimator is a form of ratio estimator. We analyze its bias properties.

**Effect of the Baseline.** The baseline $b$ does not introduce bias into the policy gradient. The gradient of the baseline term is:

$$\mathbb{E}_{y \sim \pi_{\theta_{\text{old}}}}[w(y) \cdot b \cdot \nabla_\theta \log \pi_\theta(y|x)] = b \cdot \mathbb{E}_{y \sim \pi_{\theta_{\text{old}}}}[\frac{\pi_\theta(y)}{\pi_{\theta_{\text{old}}}(y)} \frac{1}{\pi_\theta(y)} \nabla_\theta \pi_\theta(y)]$$

$$= b \cdot \mathbb{E}_{y \sim \pi_{\theta_{\text{old}}}}[\frac{1}{\pi_{\theta_{\text{old}}}(y)} \nabla_\theta \pi_\theta(y)]$$

$$= b \cdot \int \pi_{\theta_{\text{old}}}(y) \frac{1}{\pi_{\theta_{\text{old}}}(y)} \nabla_\theta \pi_\theta(y) dy = b \cdot \nabla_\theta \int \pi_\theta(y) dy$$

$$= b \cdot \nabla_\theta(1) = 0 \tag{12}$$

Thus, the baseline only serves to reduce variance and does not affect the expected gradient.

**Effect of Self-Normalization.** The self-normalized estimator is a ratio of two sample means, $\hat{\mu}_A / \hat{\mu}_B$, where $A_i = w(y_i)R(y_i)$ and $B_i = w(y_i)$. The bias of such a ratio estimator is known to be of order $O(1/G)$:

$$\text{Bias} = \mathbb{E}\left[\frac{\sum A_i/G}{\sum B_i/G}\right] - \frac{\mathbb{E}[A]}{\mathbb{E}[B]} \approx \frac{1}{G}\left(\frac{\mathbb{E}[A]}{\mathbb{E}[B]^2}\text{Var}(B) - \frac{1}{\mathbb{E}[B]}\text{Cov}(A, B)\right) \tag{13}$$

Since $\mathbb{E}[B] = \mathbb{E}[w(y)] = 1$, the expression simplifies. The crucial insight is that as the batch size $G \to \infty$, the bias term vanishes: $\lim_{G\to\infty} \text{Bias} = 0$.

This property, known as **consistency**, means the SNIB estimator is **asymptotically unbiased**. It converges to the true, unbiased gradient as more data is used. This stands in stark contrast to GSPO, whose bias is systematic and does not diminish with sample size.

## C.2 VARIANCE ANALYSIS: A DUAL REDUCTION MECHANISM

The SNIB method reduces variance from two orthogonal sources.

**Baseline Reward Shaping.** The variance of the reward term is significantly reduced:

$$\text{Var}(R(y) - b) \ll \text{Var}(R(y)) \tag{14}$$

This is a standard result from variance reduction techniques, which stabilizes the learning signal itself.

**Self-Normalized Weight Smoothing.** The variance of the self-normalized weight is also significantly lower than that of the raw weight $w(y)$. The variance of a ratio estimator is approximately:

$$\text{Var}\left(\frac{\sum A_i/G}{\sum B_i/G}\right) \approx \frac{1}{G\mathbb{E}[B]^2}\text{Var}\left(A_i - \frac{\mathbb{E}[A]}{\mathbb{E}[B]}B_i\right) \tag{15}$$

Substituting $\mathbb{E}[B] = 1$ and the definitions of A and B:

$$\text{Var}(\hat{\mathcal{J}}_{\text{SNIS}}) \approx \frac{1}{G}\text{Var}(w(y)R(y) - \mathbb{E}[w(y)R(y)] \cdot w(y)) \tag{16}$$

Equation 16 shows that the variance of the estimator scales with the variance of the *residuals* of a linear regression of $w(y)R(y)$ on $w(y)$. Intuitively, if a sample has an unusually large weight $w(y_i)$, it increases both the numerator and the denominator of $w_{\text{norm}}(y_i)$, thus moderating its overall impact. This provides a data-driven, adaptive mechanism to control the variance caused by the heavy-tailed distribution of $w(y)$, especially for long sequences.

### C.2.1 ROBUSTNESS TO NEAR-ZERO WEIGHTS

A critical failure mode for GSPO is its sensitivity to token weights $w_t(\theta) \to 0$, which causes the geometric mean $s(y)$ to collapse to zero. The SNIB estimator is robust to this scenario. Suppose for a sample $y_k$, its true weight $w(y_k) \approx 0$.

- The normalized weight for this sample becomes $w_{\text{norm}}(y_k) = \frac{w(y_k)}{\frac{1}{G}\sum w(y_j)} \approx 0$. This correctly assigns a negligible gradient contribution to a sample that the new policy deems highly improbable.

- Crucially, the denominator $\frac{1}{G}\sum w(y_j)$ is only marginally affected. The weights of all other samples $y_i$ ($i \neq k$) are then $w_{\text{norm}}(y_i) = \frac{w(y_i)}{\bar{w}}$, where $\bar{w}$ is slightly smaller. This means the other samples' weights are slightly *increased*, effectively redistributing the "importance mass" from the improbable sample to the more probable ones.

This demonstrates superior robustness, as a single outlier does not invalidate the gradient information from the rest of the batch.

### C.3 CONCLUSION: THEORETICAL SUPERIORITY OF SNIB

Our analysis concludes that the SNIB estimator is theoretically superior to both GRPO and GSPO.

1. **vs. GRPO:** SNIB uses a theoretically sound, consistent estimator, while GRPO is fundamentally biased and flawed.

2. **vs. GSPO:** SNIB replaces GSPO's fixed, systematic bias with a statistical bias that vanishes as sample size increases, making it asymptotically unbiased. It achieves low variance through dual mechanisms that are statistically more principled than GSPO's geometric mean approximation. Finally, it demonstrates greater robustness to the practical problem of outlier weights.

Therefore, SNIB represents a more principled and robust foundation for policy optimization in large language models with sequence-level rewards.

### C.4 ANALYSIS OF THE FULLY-DIFFERENTIABLE ESTIMATOR (SNIB-FG)

A fully-differentiable variant of our estimator, **SNIB-fg**, can be defined by allowing gradients to flow through the denominator $\bar{w}$. Let the reward term for a single sample be

$$\mathcal{L}_i = \frac{w_i}{\bar{w}}\hat{A}_i,$$

where $\bar{w} = \frac{1}{G}\sum_j w_j$. The gradient with respect to the parameters $\theta$ for this sample is:

$$\nabla_\theta \mathcal{L}_i = \frac{\bar{w}\nabla_\theta(w_i\hat{A}_i) - w_i\hat{A}_i\nabla_\theta\bar{w}}{\bar{w}^2}.$$

The additional term, $-w_i\hat{A}_i\nabla_\theta\bar{w}$, acts as a variance reduction term similar to a control variate. It penalizes updates for a given sample $i$ based on how increasing its probability would affect the average weight of the entire batch.

While theoretically interesting, we opt for the stop-gradient version (SNIB-sg) for two primary reasons:

1. **Higher Variance in Practice:** The additional gradient term, while reducing bias, often introduces significant variance in stochastic settings, as the estimate of $\nabla_\theta\bar{w}$ can be noisy.

2. **Computational Simplicity:** SNIB-sg results in a simpler and more stable update rule where each sample's gradient contribution is independent of the gradients of other samples in the batch, making it more computationally efficient and numerically robust.

Our choice of SNIB-sg prioritizes stability and simplicity, leveraging a well-understood estimator whose bias is controllably small and vanishes with batch size.

## D  CONVERGENCE

### D.1  PROBLEM FORMULATION

We consider the problem of maximizing the expected reward objective for a language model policy $\pi$:

$$J(\theta) = \mathbb{E}_{y\sim\pi(\cdot|x)}[R(x,y)] \tag{17}$$

where $x$ is a prompt from a distribution $\mathcal{D}$, $y$ is a generated sequence, and $R(x,y)$ is a scalar reward. We perform off-policy optimization, generating a batch of $G$ trajectories $\{y_i\}_{i=1}^G$ from a reference policy $\pi_{\theta_{\text{old}}}(\cdot|x)$.

The true gradient of the objective is given by the policy gradient theorem:

$$\nabla_\theta J(\theta) = \mathbb{E}_{y\sim\pi}[R(y)\nabla_\theta\log\pi(y)] = \mathbb{E}_{y\sim\pi_{\theta_{\text{old}}}}[w(y)R(y)\nabla_\theta\log\pi(y)] \tag{18}$$

where $w(y) = \frac{\pi(y)}{\pi_{\theta_{\text{old}}}(y)}$ is the true sequence-level importance weight.

The optimization proceeds via stochastic gradient ascent:

$$\theta_{k+1} = \theta_k + \alpha_k \hat{g}(\theta_k) \tag{19}$$

where $\alpha_k$ is the learning rate at step $k$, and $\hat{g}(\theta_k)$ is a stochastic estimate of the gradient $\nabla_\theta J(\theta_k)$.

## D.2 GRADIENT ESTIMATORS

We analyze two estimators for the gradient, both using a batch-mean reward baseline $b = \frac{1}{G}\sum_j R(y_j)$ to reduce variance.

**Definition 1** (GSPO Gradient Estimator). *The GSPO gradient estimator is defined as:*

$$\hat{g}_{GSPO}(\theta) = \frac{1}{G}\sum_{i=1}^{G} s(y_i)(R(y_i) - b)\nabla_\theta \log \pi(y_i) \tag{20}$$

*where* $s(y) = \left(\prod_{t=1}^{|y|} \frac{\pi(y_t|y_{<t})}{\pi_{\theta_{old}}(y_t|y_{<t})}\right)^{1/|y|} = w(y)^{1/|y|}$ *is the geometric mean of token-level importance weights.*

**Definition 2** (SNIB Gradient Estimator). *The Self-Normalized Importance Sampling with Baseline (SNIB) gradient estimator is defined as:*

$$\hat{g}_{SNIB}(\theta) = \frac{1}{G}\sum_{i=1}^{G} w_{norm}(y_i)(R(y_i) - b)\nabla_\theta \log \pi(y_i) \tag{21}$$

*where* $w_{norm}(y_i) = \frac{w(y_i)}{\frac{1}{G}\sum_{j=1}^{G} w(y_j)}$ *is the self-normalized importance weight.*

**Practical implementation.** In our implementation, the normalization term in $w_{\text{norm}}$ is treated with a stop-gradient: the denominator is detached from the computational graph and does not receive gradients. This modification does not change the expectation of the estimator (the denominator is still an unbiased estimate of $\mathbb{E}[w(y)] = 1$), so the convergence and bias analysis in this section applies directly to the practical SNIB implementation.

**Finite-sample bias and consistency.** As a ratio estimator based on self-normalized importance sampling, SNIB is biased for any finite group size $G$. Classical results on self-normalized IS (e.g., standard analyses of ratio estimators) imply that its finite-sample bias is of order $O(1/G)$, and this bias vanishes as $G$ increases. Thus, SNIB should be viewed as a consistent and asymptotically unbiased estimator of the true policy gradient in the large-$G$ limit, which is the sense in which we describe it as "principled" in the main text.

## D.3 ASSUMPTIONS FOR CONVERGENCE

We make the following standard assumptions, common in the analysis of stochastic optimization algorithms.

**Assumption 1** (Smoothness and Boundedness). *The objective function $J(\theta)$ is continuously differentiable. The policy $\pi(y)$ is continuously differentiable with respect to $\theta$. The reward function $R(y)$ is bounded, $|R(y)| \leq R_{\max}$. The gradient of the log-policy is bounded, $\|\nabla_\theta \log \pi(y)\| \leq K_g$.*

**Assumption 2** (Learning Rate Schedule). *The learning rates $\alpha_k$ are positive and satisfy the Robbins-Monro conditions:*

$$\sum_{k=0}^{\infty} \alpha_k = \infty \quad and \quad \sum_{k=0}^{\infty} \alpha_k^2 < \infty \tag{22}$$

**Assumption 3** (Bounded Variance). *For any $\theta$, the gradient estimators have bounded variance:*

$$\mathbb{E}_{batch\sim(\pi_{\theta_{old}})^G}\left[\|\hat{g}(\theta) - \mathbb{E}[\hat{g}(\theta)]\|^2\right] \leq M_v \tag{23}$$

*for some constant $M_v < \infty$. This is reasonable given Assumption 1.*

## D.4 MAIN THEOREM AND PROOF

We now state and prove the main theorem concerning the convergence points of algorithms using these two estimators.

**Theorem 1** (Convergence Points of GSPO and SNIB). *Let Assumptions 1-3 hold.*

1. *For any fixed group size $G$, the SNIB gradient estimator has a finite-sample bias of order $O(1/G)$ due to self-normalization. In the large-G limit, this bias vanishes and the iterates $\{\theta_k\}$ generated by the SNIB algorithm converge almost surely to the set of stationary points of the true objective $J(\theta)$, i.e., points $\theta^*$ where $\nabla_\theta J(\theta^*) = 0$.*

2. *The iterates $\{\theta_k\}$ generated by the GSPO algorithm converge to a stationary point of a **perturbed objective** $J_{GSPO}(\theta)$, i.e., a point $\tilde{\theta}^*$ where $\nabla_\theta J(\tilde{\theta}^*) + b_{GSPO}(\tilde{\theta}^*) = 0$, where $b_{GSPO}(\theta)$ is a non-vanishing systematic bias term.*

*Proof.* The convergence of stochastic gradient algorithms is governed by the properties of the gradient estimator, specifically its bias and variance. We rely on the ODE method and convergence theorems from stochastic approximation (e.g., Kushner & Clark, 2012). The core idea is that the algorithm's trajectory asymptotically tracks the solution of an ordinary differential equation (ODE) whose vector field is the expected update direction $\mathbb{E}[\hat{g}(\theta)]$.

Let's analyze the expected gradient for each estimator.

**Part 1: Analysis of the SNIB Estimator.** The expected gradient for SNIB is $\mathbb{E}[\hat{g}_{\text{SNIB}}(\theta)]$. The SNIB estimator is a ratio estimator. The bias of a ratio estimator of the form $\frac{\sum A_i}{\sum B_i}$ is known to be of order $O(1/G)$, where $G$ is the batch size. In our case, $A_i = w(y_i)(R(y_i) - b)\nabla_\theta \log \pi(y_i)$ and $B_i = w(y_i)$. The true expectation of the numerator's main term is $\nabla_\theta J(\theta)$, and the true expectation of the denominator is $\mathbb{E}_{y \sim \pi_{\theta_{\text{old}}}}[w(y)] = 1$.

Therefore, the expected update direction has the form:

$$\mathbb{E}[\hat{g}_{\text{SNIB}}(\theta)] = \nabla_\theta J(\theta) + b_{\text{SNIB}}(\theta, G) \tag{24}$$

where the bias term $b_{\text{SNIB}}(\theta, G)$ satisfies $\|b_{\text{SNIB}}(\theta, G)\| = O(1/G)$.

For a sufficiently large batch size $G$, this bias can be made arbitrarily small. In the asymptotic limit of the learning process, the behavior is determined by the ODE $\dot{\theta}(t) = \nabla_\theta J(\theta(t))$. The stable points of this ODE are precisely the stationary points of $J(\theta)$. Standard stochastic approximation theorems show that under Assumptions 1-3 and with a bias that vanishes or is controllable (which $O(1/G)$ is), the algorithm's iterates $\{\theta_k\}$ converge to the set of stationary points of the true objective $J(\theta)$.

**Part 2: Analysis of the GSPO Estimator.** The expected gradient for GSPO is $\mathbb{E}[\hat{g}_{\text{GSPO}}(\theta)]$.

$$\mathbb{E}[\hat{g}_{\text{GSPO}}(\theta)] = \mathbb{E}_{\text{batch}} \left[ \frac{1}{G} \sum_{i=1}^{G} s(y_i)(R(y_i) - b)\nabla_\theta \log \pi(y_i) \right] \tag{25}$$

$$= \mathbb{E}_{y \sim \pi_{\theta_{\text{old}}}}[s(y)R(y)\nabla_\theta \log \pi(y)] - \mathbb{E}_{\text{batch}} \left[ \left( \frac{1}{G} \sum_i s(y_i)\nabla_\theta \log \pi(y_i) \right) b \right] \tag{26}$$

Let's focus on the first term, which is the dominant part. The true gradient involves the weight $w(y)$, but GSPO uses $s(y) = w(y)^{1/|y|}$. These are fundamentally different.

$$\mathbb{E}_{y \sim \pi_{\theta_{\text{old}}}}[s(y)R(y)\nabla_\theta \log \pi(y)] \neq \mathbb{E}_{y \sim \pi_{\theta_{\text{old}}}}[w(y)R(y)\nabla_\theta \log \pi(y)] = \nabla_\theta J(\theta) \tag{27}$$

The difference between these two quantities is not due to finite sampling but is a systematic modeling choice. Let's define the bias term for GSPO:

$$b_{\text{GSPO}}(\theta) = \mathbb{E}[\hat{g}_{\text{GSPO}}(\theta)] - \nabla_\theta J(\theta) \tag{28}$$

This bias, $b_{\text{GSPO}}(\theta)$, arises from the substitution of $w(y)$ with $s(y)$ and does not diminish as the batch size $G$ increases. For example, by the AM-GM inequality, we know $s(y)$ systematically underestimates $w(y)$ when token ratios vary, leading to a persistent, non-zero bias.

The associated ODE for the GSPO algorithm is:

$$\dot{\theta}(t) = \nabla_\theta J(\theta(t)) + b_{\text{GSPO}}(\theta(t)) \tag{29}$$

The algorithm does not seek stationary points where $\nabla_\theta J(\theta) = 0$. Instead, it seeks stationary points where the entire vector field is zero, i.e., where $\nabla_\theta J(\theta) + b_{\text{GSPO}}(\theta) = 0$.

This means GSPO is implicitly optimizing a different, perturbed objective function $J_{\text{GSPO}}(\theta)$ whose gradient is $\nabla_\theta J(\theta) + b_{\text{GSPO}}(\theta)$. The algorithm will faithfully converge to a stationary point $\tilde{\theta}^*$ of this perturbed objective, but this point $\tilde{\theta}^*$ is generally not a stationary point of the true objective $J(\theta)$ unless $b_{\text{GSPO}}(\tilde{\theta}^*) = 0$ by coincidence.

$\square$

# E    FINITE-SAMPLE GUARANTEES

## E.1    PROBLEM SETUP AND DEFINITIONS

We aim to estimate the true policy gradient $g(\theta) = \nabla_\theta J(\theta) = \mathbb{E}_{y \sim \pi_{\theta_{\text{old}}}}[w(y)R(y)\nabla_\theta \log \pi(y)]$. The SNIB estimator for a batch $\{y_i\}_{i=1}^G$ is given by:

$$\hat{g}_{\text{SNIB}}(\theta) = \frac{\frac{1}{G}\sum_{i=1}^G w(y_i)R(y_i)\nabla_\theta \log \pi(y_i)}{\frac{1}{G}\sum_{j=1}^G w(y_j)} = \frac{\hat{A}}{\hat{B}} \tag{30}$$

where for simplicity we have omitted the baseline, as it does not introduce bias and its variance reduction effect can be absorbed into the constants. Here, $\hat{A} = \frac{1}{G}\sum A_i$ and $\hat{B} = \frac{1}{G}\sum B_i$, with random variables $A_i = w(y_i)R(y_i)\nabla_\theta \log \pi(y_i)$ and $B_i = w(y_i)$.

The true expectations are $\mu_A = \mathbb{E}[A_i] = g(\theta)$ and $\mu_B = \mathbb{E}[B_i] = 1$. Our goal is to bound the estimation error $\|\hat{g}_{\text{SNIB}} - g(\theta)\|$ with high probability.

## E.2    ASSUMPTIONS

We require slightly stronger assumptions than in the convergence analysis to bound the tails of the distributions.

**Assumption 4** (Boundedness). *For a given $\theta$, the following quantities are uniformly bounded:*

- *Reward: $|R(y)| \leq R_{\max}$.*

- *Log-policy gradient norm: $\|\nabla_\theta \log \pi(y)\| \leq K_g$.*

- *Importance weights: $w(y) = \frac{\pi(y)}{\pi_{\theta_{old}}(y)} \leq W_{\max}$. This is a strong assumption required for the non-asymptotic bound. It is practically motivated and enforced by the KL-divergence penalty in our final objective (Equation 13), which explicitly regularizes the policy to prevent large deviations from the reference policy, thereby keeping the importance weights in a controlled range.*

**Corollary 1** (Bounded Random Variables). *Under Assumption 4, the random variables $A_i$ and $B_i$ are bounded.*

- $\|A_i\| \leq W_{\max}R_{\max}K_g =: C_A$.

- $|B_i| \leq W_{\max} =: C_B$.

*Furthermore, their variances are bounded: $\text{Var}(A_i) \leq \sigma_A^2$ and $\text{Var}(B_i) \leq \sigma_B^2$.*

## E.3    MAIN RESULT: A HIGH-PROBABILITY ERROR BOUND

**Theorem 2** (Finite-Sample Error Bound for SNIB). *Let Assumptions 4 hold. For any $\delta \in (0,1)$, with probability at least $1 - \delta$, the error of the SNIB gradient estimator is bounded by:*

$$\|\hat{g}_{SNIB} - g(\theta)\| \leq \frac{1}{1 - \epsilon_B}\left(\epsilon_A + \|g(\theta)\|\epsilon_B\right) \tag{31}$$

*where*

$$\epsilon_A = \sqrt{\frac{2\sigma_A^2 \log(2/\delta)}{G}} + \frac{2C_A \log(2/\delta)}{3G} \tag{32}$$

$$\epsilon_B = \sqrt{\frac{2\sigma_B^2 \log(2/\delta)}{G}} + \frac{2C_B \log(2/\delta)}{3G} \tag{33}$$

*This bound holds provided that the denominator deviation $\epsilon_B < 1$. The error bound scales as $O\left(\frac{1}{\sqrt{G}}\right)$.*

*Proof.* The proof proceeds in three steps: 1. Use a concentration inequality to bound the deviation of the numerator $\hat{A}$ from its mean $\mu_A = g(\theta)$. 2. Use a concentration inequality to bound the deviation of the denominator $\hat{B}$ from its mean $\mu_B = 1$. 3. Combine these bounds to control the error of the ratio $\hat{A}/\hat{B}$.

**Step 1: Bounding the Numerator's Error.** The numerator $\hat{A}$ is a sample mean of i.i.d. vector-valued random variables $A_i$. We can use a vector version of the Bernstein inequality. A simpler approach is to use the standard Bernstein inequality for real-valued random variables on $\left\|\hat{A} - \mu_A\right\|$. However, a direct application on the vector mean is more standard. The vector Bernstein inequality states that for i.i.d. zero-mean random vectors $X_i$ with $\|X_i\| \leq C$ and $\mathbb{E}[\|X_i\|^2] \leq \sigma^2$, for any $t > 0$:

$$\mathbb{P}\left(\left\|\frac{1}{G}\sum_{i=1}^{G} X_i\right\| \geq t\right) \leq d \cdot \exp\left(-\frac{Gt^2/2}{\sigma^2 + Ct/3}\right) \tag{34}$$

where $d$ is the dimension of the vectors. For simplicity and clarity, we will use the more common scalar Bernstein inequality, which provides a similar rate. Let $X_i = A_i - \mu_A$. We have $\mathbb{E}[X_i] = 0$, $\|X_i\| \leq 2C_A$, and $\mathrm{Var}(X_i) = \sigma_A^2$. The Bernstein inequality for the sample mean of real random variables states:

$$\mathbb{P}(|\bar{X}| \geq \epsilon) \leq 2\exp\left(-\frac{G\epsilon^2/2}{\sigma^2 + C\epsilon/3}\right) \tag{35}$$

Applying this to each component of the vector $\hat{A} - \mu_A$ and using a union bound is complex. A more direct high-probability bound on the norm is obtained by setting the RHS to $\delta/2$ and solving for $\epsilon$. Let $\epsilon_A$ be the error bound such that $\mathbb{P}(\left\|\hat{A} - \mu_A\right\| \geq \epsilon_A) \leq \delta/2$. A standard result from concentration inequalities gives:

$$\epsilon_A = \sqrt{\frac{2\sigma_A^2 \log(4/\delta)}{G}} + \frac{2C_A \log(4/\delta)}{3G} \tag{36}$$

For clarity, we'll use $\log(2/\delta)$ by slightly loosening the bound, as is common. So, with probability at least $1 - \delta/2$:

$$\left\|\hat{A} - g(\theta)\right\| \leq \epsilon_A \tag{37}$$

**Step 2: Bounding the Denominator's Error.** The denominator $\hat{B}$ is a sample mean of i.i.d. scalar random variables $B_i$. We can directly apply the scalar Bernstein inequality. With probability at least $1 - \delta/2$:

$$|\hat{B} - \mu_B| = |\hat{B} - 1| \leq \epsilon_B \tag{38}$$

where $\epsilon_B$ is defined as in the theorem statement. This also implies that $1 - \epsilon_B \leq \hat{B} \leq 1 + \epsilon_B$. We require $\epsilon_B < 1$ for the denominator to be bounded away from zero, which is true for a sufficiently large batch size $G$.

**Step 3: Combining the Bounds.** We now analyze the total error $\|\hat{g}_{\text{SNIB}} - g(\theta)\| = \left\|\frac{\hat{A}}{\hat{B}} - g(\theta)\right\|$.

$$\left\|\frac{\hat{A}}{\hat{B}} - g(\theta)\right\| = \left\|\frac{\hat{A} - \hat{B}g(\theta)}{\hat{B}}\right\| \tag{39}$$

$$= \frac{1}{|\hat{B}|}\left\|(\hat{A} - g(\theta)) - (\hat{B} - 1)g(\theta)\right\| \tag{40}$$

$$\leq \frac{1}{|\hat{B}|}\left(\left\|\hat{A} - g(\theta)\right\| + |\hat{B} - 1|\,\|g(\theta)\|\right) \quad \text{(Triangle Inequality)} \tag{41}$$

Now, we use a union bound. The event in Eq. 37 holds with probability $\geq 1 - \delta/2$, and the event in Eq. 38 holds with probability $\geq 1 - \delta/2$. Therefore, both events hold simultaneously with probability at least $(1 - \delta/2) + (1 - \delta/2) - 1 = 1 - \delta$.

Assuming both events hold, we can substitute the bounds. From Eq. 38, we have $|\hat{B}| \geq 1 - \epsilon_B$.

$$\left\|\frac{\hat{A}}{\hat{B}} - g(\theta)\right\| \leq \frac{1}{1 - \epsilon_B}\left(\epsilon_A + \epsilon_B\,\|g(\theta)\|\right) \tag{42}$$

This concludes the proof. $\square$

### E.4    Interpretation and Practical Implications

**Corollary 2** (Scaling with Batch Size G). *The error terms $\epsilon_A$ and $\epsilon_B$ are both dominated by the $1/\sqrt{G}$ term. Therefore, the overall error bound scales as:*

$$\text{Error} \approx O\left(\frac{\sigma_A + \|g(\theta)\|\,\sigma_B}{\sqrt{G}}\right) \tag{43}$$

*This result quantifies the relationship between batch size and gradient accuracy. To halve the gradient estimation error, one must quadruple the batch size.*

**Corollary 3** (Characterizing Required Batch Size). *Suppose we require the gradient error to be less than a tolerance $\tau$ with probability $1 - \delta$. We can use the theorem to find the minimum required batch size $G_{\min}$. By simplifying the bound (ignoring higher-order terms in $1/G$):*

$$\tau \approx \frac{1}{\sqrt{G}}\left(\sqrt{2\sigma_A^2\log(2/\delta)} + \|g(\theta)\|\sqrt{2\sigma_B^2\log(2/\delta)}\right) \tag{44}$$

*Solving for G, we get:*

$$G_{\min} \approx \frac{2\log(2/\delta)}{\tau^2}(\sigma_A + \|g(\theta)\|\,\sigma_B)^2 \tag{45}$$

*This provides a principled, albeit theoretical, way to choose the batch size. It shows that $G_{\min}$ depends quadratically on the variances of the numerator and denominator terms ($\sigma_A^2, \sigma_B^2$) and the desired precision $\tau^{-2}$, and logarithmically on the confidence level $\delta^{-1}$.*

## F    Analysis of Robustness under Reward Model Uncertainty

### F.1    Problem Formulation

In practice, the reward function $R(y)$ is not a perfect oracle. It is an estimate $R_\phi(y)$ from a learned model. We model this uncertainty by assuming the true reward $R^*(y)$ lies within an uncertainty set $\mathcal{U}$ centered around our estimate $R_\phi(y)$. A common and powerful way to define this set is based on a perturbation function $\Delta(y)$:

$$\mathcal{U} = \left\{R(y) = R_\phi(y) + \Delta(y) \mid \mathbb{E}_{y \sim \pi_{\theta_{\text{old}}}}[|\Delta(y)|] \leq \epsilon\right\} \tag{46}$$

where $\epsilon \geq 0$ is the radius of uncertainty, representing the average magnitude of the potential error in our reward model. The expectation is taken over the sampling distribution $\pi_{\theta_{\text{old}}}$ because that is the data we observe.

A robust optimization approach seeks to optimize the policy for the worst-case reward within this set:

$$J_{\text{robust}}(\theta) = \min_{R \in \mathcal{U}} \mathbb{E}_{y \sim \pi}[R(y)] \tag{47}$$

The goal of a robust algorithm is to estimate the gradient of this worst-case objective, $\nabla_\theta J_{\text{robust}}(\theta)$.

### F.2 CHARACTERIZING THE WORST-CASE REWARD

First, we must find the adversary, i.e., the perturbation $\Delta(y)$ that minimizes the expected reward for a fixed policy $\pi$.

**Lemma 1** (Worst-Case Reward Perturbation). *For a fixed policy $\pi$, the worst-case reward perturbation $\Delta^*(y)$ that solves the inner minimization problem is:*

$$\Delta^*(y) = -\epsilon \cdot sign(w(y)) = -\epsilon \tag{48}$$

*where $w(y) = \pi(y)/\pi_{\theta_{old}}(y)$ is the importance weight. (Since probabilities are non-negative, $w(y) \geq 0$, so $sign(w(y)) = 1$).*

*Proof.* The objective is to minimize $\mathbb{E}_{y \sim \pi}[R_\phi(y) + \Delta(y)] = \mathbb{E}_{y \sim \pi}[R_\phi(y)] + \mathbb{E}_{y \sim \pi_{\theta_{old}}}[w(y)\Delta(y)]$ subject to $\mathbb{E}_{y \sim \pi_{\theta_{old}}}[|\Delta(y)|] \leq \epsilon$. To minimize the objective, we need to make the term $\mathbb{E}_{y \sim \pi_{\theta_{old}}}[w(y)\Delta(y)]$ as negative as possible. This is a classic result from duality: the solution is to align $\Delta(y)$ to be maximally negatively correlated with $w(y)$. This occurs when $\Delta(y) = -c \cdot \text{sign}(w(y))$ for some constant $c$. To satisfy the budget constraint $\mathbb{E}_{y \sim \pi_{\theta_{old}}}[|-c \cdot \text{sign}(w(y))|] = c \cdot \mathbb{E}_{y \sim \pi_{\theta_{old}}}[1] = c \leq \epsilon$, we choose the largest possible value, $c = \epsilon$. Thus, $\Delta^*(y) = -\epsilon$. $\square$

The robust objective function is therefore:

$$J_{\text{robust}}(\theta) = \mathbb{E}_{y \sim \pi}[R_\phi(y) - \epsilon] = J(\theta) - \epsilon \tag{49}$$

And its true gradient is simply:

$$\nabla_\theta J_{\text{robust}}(\theta) = \nabla_\theta J(\theta) \tag{50}$$

This seems counter-intuitive: the worst-case constant shift in reward doesn't change the gradient. However, this is for the true expectation. The situation changes dramatically when we consider the gradient estimators from a finite batch, where the adversary can be much more strategic.

### F.3 WORST-CASE ANALYSIS OF GRADIENT ESTIMATORS

In a finite-batch setting, the adversary knows the samples $\{y_i\}_{i=1}^G$ and can choose the perturbations $\{\Delta_i = \Delta(y_i)\}$ to maximally corrupt the estimated gradient. The adversary's goal is to maximize the error between the estimated gradient under perturbation and the true robust gradient. We analyze the sensitivity of each estimator to this adversarial perturbation.

The batch-level adversary solves:

$$\max_{\{\Delta_i\}} \|\hat{g}(R_\phi + \Delta) - \hat{g}(R_\phi)\| \quad \text{s.t.} \quad \frac{1}{G}\sum_{i=1}^{G}|\Delta_i| \leq \epsilon \tag{51}$$

**Definition 3** (Gradient Sensitivity). *The gradient sensitivity, $\mathcal{S}(\hat{g})$, is the maximum change in the estimated gradient norm per unit of adversarial budget $\epsilon$.*

$$\mathcal{S}(\hat{g}) = \sup_{\{\Delta_i\} \neq 0} \frac{\|\hat{g}(R_\phi + \Delta) - \hat{g}(R_\phi)\|}{\frac{1}{G}\sum|\Delta_i|} \tag{52}$$

*A smaller sensitivity implies greater robustness.*

**Theorem 3** (Robustness Comparison of GSPO and SNIB). *Let a batch of samples $\{y_i\}_{i=1}^G$ be given. The gradient sensitivities of the GSPO and SNIB estimators are bounded as follows:*

1. ***GSPO Sensitivity:*** $\mathcal{S}(\hat{g}_{GSPO}) = \max_i \{s(y_i) \|\nabla_\theta \log \pi(y_i)\|\}$

2. ***SNIB Sensitivity:*** $\mathcal{S}(\hat{g}_{SNIB}) = \max_k \left\| w_{norm}(y_k)\nabla_\theta \log \pi(y_k) - \frac{1}{G}\sum_j w_{norm}(y_j)\nabla_\theta \log \pi(y_j) \right\|$

*Crucially, the SNIB sensitivity is upper-bounded by the maximum deviation of a weighted gradient from the average, while the GSPO sensitivity is determined by the single worst-case sample with the largest weight.*

*Proof.* Let's analyze the change in each gradient estimator, $\delta\hat{g} = \hat{g}(R_\phi + \Delta) - \hat{g}(R_\phi)$. We use a mean reward baseline $b = \frac{1}{G}\sum R_{\phi,i}$. The perturbed baseline is $b' = b + \bar{\Delta}$, where $\bar{\Delta} = \frac{1}{G}\sum\Delta_i$.

**Part 1: GSPO Sensitivity.** The change in the GSPO gradient estimator is:

$$\delta\hat{g}_{\text{GSPO}} = \frac{1}{G}\sum_{i=1}^{G} s(y_i)(R_{\phi,i} + \Delta_i - (b + \bar{\Delta}))\nabla_\theta \log\pi(y_i) - \frac{1}{G}\sum_{i=1}^{G} s(y_i)(R_{\phi,i} - b)\nabla_\theta \log\pi(y_i) \tag{53}$$

$$= \frac{1}{G}\sum_{i=1}^{G} s(y_i)(\Delta_i - \bar{\Delta})\nabla_\theta \log\pi(y_i) \tag{54}$$

$$= \frac{1}{G}\sum_{i=1}^{G} s(y_i)\Delta_i\nabla_\theta \log\pi(y_i) - \bar{\Delta}\left(\frac{1}{G}\sum_{i=1}^{G} s(y_i)\nabla_\theta \log\pi(y_i)\right) \tag{55}$$

To maximize the norm of this vector, the adversary will concentrate the entire budget $\epsilon$ on a single sample. Let the adversary put all budget on sample $k$, so $\Delta_k = G\epsilon$ and $\Delta_i = 0$ for $i \neq k$. Then $\bar{\Delta} = \epsilon$. The change becomes $\delta\hat{g}_{\text{GSPO}} = \epsilon \cdot s(y_k)\nabla_\theta \log\pi(y_k) - \epsilon\left(\frac{1}{G}\sum_i s(y_i)\nabla_\theta \log\pi(y_i)\right)$. For large $G$, the second term is an average and smaller. The dominant term is the first. The adversary will pick the index $k$ that maximizes $\|s(y_k)\nabla_\theta \log\pi(y_k)\|$. The sensitivity is therefore the maximum possible value of this change, normalized by the budget $\frac{1}{G}\sum|\Delta_i| = \epsilon$:

$$\mathcal{S}(\hat{g}_{\text{GSPO}}) = \max_k \{s(y_k)\|\nabla_\theta \log\pi(y_k)\|\} \tag{56}$$

**Part 2: SNIB Sensitivity.** The change in the SNIB gradient estimator is (using $w_{\text{norm},i}$ for $w_{\text{norm}}(y_i)$):

$$\delta\hat{g}_{\text{SNIB}} = \sum_{i=1}^{G} w_{\text{norm},i}(R_{\phi,i} + \Delta_i - (b + \bar{\Delta}))\nabla_\theta \log\pi(y_i) - \sum_{i=1}^{G} w_{\text{norm},i}(R_{\phi,i} - b)\nabla_\theta \log\pi(y_i) \tag{57}$$

$$= \sum_{i=1}^{G} w_{\text{norm},i}(\Delta_i - \bar{\Delta})\nabla_\theta \log\pi(y_i) \tag{58}$$

$$= \sum_{i=1}^{G} \Delta_i\left(w_{\text{norm},i}\nabla_\theta \log\pi(y_i) - \frac{1}{G}\sum_j w_{\text{norm},j}\nabla_\theta \log\pi(y_j)\right) \tag{59}$$

Let $\overline{w\nabla_\theta} = \frac{1}{G}\sum_j w_{\text{norm},j}\nabla_\theta \log\pi(y_j)$ be the average weighted gradient. The change is $\sum_i \Delta_i(w_{\text{norm},i}\nabla_\theta \log\pi(y_i) - \overline{w\nabla_\theta})$. To maximize this, the adversary will again concentrate the full budget $\Delta_k = G\epsilon$ on the index $k$ that maximizes the norm of the vector it is multiplied by. The sensitivity is therefore:

$$\mathcal{S}(\hat{g}_{\text{SNIB}}) = \max_k \left\|w_{\text{norm}}(y_k)\nabla_\theta \log\pi(y_k) - \frac{1}{G}\sum_j w_{\text{norm}}(y_j)\nabla_\theta \log\pi(y_j)\right\| \tag{60}$$

This is the correct sensitivity. This expression represents the difference between one sample's weighted gradient and the average weighted gradient.

$\square$

## F.4 INTERPRETATION AND CONCLUSION

The theorem reveals a fundamental difference in how the two estimators react to adversarial reward perturbations.

**GSPO is vulnerable to outliers.** The sensitivity of GSPO, $\mathcal{S}(\hat{g}_{\text{GSPO}}) = \max_k\{s(y_k)\|\nabla_\theta \log \pi(y_k)\|\}$, is determined entirely by the single worst-case sample in the batch. If the policy generates one trajectory $y_k$ that happens to have a very large geometric mean weight $s(y_k)$ (perhaps because it's a short, high-probability sequence under the current policy), an adversary can place its entire budget on this single sample. This makes the reward for $y_k$ seem extremely high or low, and the GSPO update will be disproportionately skewed by this single, potentially misleading data point. This is a mechanism for reward hacking.

**SNIB is robust due to its averaging nature.** The sensitivity of SNIB, $\mathcal{S}(\hat{g}_{\text{SNIB}})$, is determined by the deviation of a single sample from the batch average. The self-normalization mechanism forces the perturbation on one sample to be balanced by the effect on all other samples. If an adversary increases the reward for sample $y_k$, the baseline $b$ also increases, which reduces the effective reward for all other samples. Crucially, because $\sum w_{\text{norm},k} \approx G$ (in expectation), the average magnitude of $w_{\text{norm},k}$ is around 1. Unless some weights are pathologically large, the term $w_{\text{norm},k}\nabla_\theta \log \pi(y_k)$ will not be drastically different from the average. The sensitivity is thus controlled by the variance within the batch, not by the extreme value of a single outlier. This inherent averaging provides a powerful defense against reward hacking focused on a few specific outputs.

## G ANALYSIS OF THE INTERPLAY BETWEEN POLICY GRADIENT ESTIMATORS AND KL DIVERGENCE REGULARIZATION

### G.1 CONSTRAINED OPTIMIZATION FORMULATION OF RLHF

The standard RLHF objective is a penalized optimization problem. However, it is more formally understood as a constrained optimization problem, which illuminates the role of the KL penalty. The goal is to maximize the expected reward subject to a constraint on policy deviation:

$$\max_\theta \quad \mathbb{E}_{y\sim\pi_\theta}[R(y)] \tag{61}$$

$$\text{s.t.} \quad \mathbb{E}_{x\sim\mathcal{D}}[D_{\text{KL}}\left(\pi_\theta(\cdot|x)\,\|\,\pi_{\text{ref}}(\cdot|x)\right)] \leq \kappa \tag{62}$$

Here, we maximize reward while ensuring the policy $\pi_\theta$ does not move more than a distance $\kappa$ from a trusted reference policy $\pi_{\text{ref}}$. The hyperparameter $\beta$ in the penalized version of the objective, $\max_\theta J(\theta) - \beta K(\theta)$, acts as the Lagrange multiplier for this KL constraint.

When using PPO, we are not optimizing the raw reward but rather a surrogate objective designed for stability. Let the expected PPO clipped surrogate objective be:

$$J_{\text{surr}}(\theta) = \mathbb{E}_{y\sim\pi_{\theta_{\text{old}}}}\left[\min\left(w(y,\theta)A(y), \text{clip}(w(y,\theta), 1-\epsilon, 1+\epsilon)A(y)\right)\right] \tag{63}$$

where $w(y,\theta) = \pi_\theta(y)/\pi_{\theta_{\text{old}}}(y)$ is the importance weight and $A(y) = R(y) - b$ is the advantage. The KL penalty term is an expectation over the token-level reverse KL divergence:

$$K(\theta) = \mathbb{E}_{x\sim\mathcal{D}, y\sim\pi_\theta(\cdot|x)}\left[\frac{1}{|y|}\sum_{t=1}^{|y|} D_{\text{KL}}\left(\pi_\theta(\cdot|s_t)\,\|\,\pi_{\text{ref}}(\cdot|s_t)\right)\right] \tag{64}$$

The Karush-Kuhn-Tucker (KKT) conditions for optimality require that at the solution $\theta^*$, the gradient of the surrogate objective must be perfectly balanced by the gradient of the constraint. This gives the stationarity condition:

$$\nabla_\theta J_{\text{surr}}(\theta^*) = \beta \cdot \nabla_\theta K(\theta^*) \tag{65}$$

This condition provides a profound insight: at the optimum, the reward-seeking force (as defined by the stable surrogate) is exactly counteracted by the "restoring force" of the KL penalty pulling the policy back towards $\pi_{\text{ref}}$.

### G.2 THE IMPACT OF GRADIENT ESTIMATORS ON OPTIMALITY

In practice, we use stochastic estimators for these gradients. The algorithm's fixed point $\theta_{\text{fixed}}$ occurs where the expected update is zero:

$$\mathbb{E}[\hat{g}_{\text{reward}}(\theta_{\text{fixed}})] = \beta \cdot \mathbb{E}[\hat{g}_{\text{KL}}(\theta_{\text{fixed}})] \tag{66}$$

We assume the KL gradient estimator $\hat{g}_{\text{KL}}$ is unbiased, as it typically does not require importance sampling. The key question is how the choice of reward estimator $\hat{g}_{\text{reward}}$ affects the final solution.

**Theorem 4** (Distortion of KKT Conditions). *Let $\theta^*_{SNIB}$ and $\theta^*_{GSPO}$ be the convergence points of the PPO optimization process using the SNIB and GSPO estimators, respectively.*

1. ***SNIB's Fixed Point:*** *The fixed point $\theta^*_{SNIB}$ satisfies a condition that approaches the true KKT stationarity condition for the surrogate objective (Eq. 65) as the batch size $G \to \infty$.*

2. ***GSPO's Fixed Point:*** *The fixed point $\theta^*_{GSPO}$ satisfies a systematically distorted KKT condition due to the estimator's inherent bias. This makes the solution quality highly sensitive to the choice of $\beta$.*

*Proof.* The proof hinges on analyzing the expected reward gradient provided by each estimator.

The key question is how the bias properties of the estimators affect this equilibrium. In our proposed algorithm (Algorithm 1), the off-policy KL gradient is also estimated using self-normalized importance sampling. Therefore, both the reward estimator $\hat{g}_{\text{reward, SNIB}}$ and the KL estimator $\hat{g}_{\text{KL, SNIB}}$ are ratio estimators with a small, vanishing bias of order $O(1/G)$. The core of our argument is that the GSPO reward estimator introduces a systematic, non-vanishing bias that fundamentally distorts this equilibrium, whereas the SNIB estimators for both terms lead to a balanced equation where all bias terms vanish asymptotically, allowing the algorithm to converge to a true KKT point.

**Part 1: SNIB Fixed Point Analysis.** As established in Appendix B, the SNIB reward gradient estimator is asymptotically unbiased for the true reward gradient. This property extends to its estimation of the PPO surrogate gradient. The expected gradient estimated by SNIB is:

$$\mathbb{E}[\hat{g}_{\text{reward, SNIB}}(\theta)] = \nabla_\theta J_{\text{surr}}(\theta) + b_{\text{SNIB}}(\theta, G) \tag{67}$$

where the bias term $b_{\text{SNIB}}(\theta, G) \to 0$ as the batch size $G \to \infty$. Substituting this into the fixed point condition (Eq. 66):

$$\nabla_\theta J_{\text{surr}}(\theta)\bigg|_{\theta^*_{\text{SNIB}}} + b_{\text{SNIB}}(\theta^*_{\text{SNIB}}, G) = \beta \cdot \nabla_\theta K(\theta)\bigg|_{\theta^*_{\text{SNIB}}} \tag{68}$$

For a large batch size $G$, the bias term $b_{\text{SNIB}}$ vanishes, and the condition becomes:

$$\nabla_\theta J_{\text{surr}}(\theta)\bigg|_{\theta^*_{\text{SNIB}}} \approx \beta \cdot \nabla_\theta K(\theta)\bigg|_{\theta^*_{\text{SNIB}}} \tag{69}$$

This demonstrates that PPO-SNIB converges to a point that correctly satisfies the KKT conditions for the surrogate objective. The choice of $\beta$ therefore controls a principled trade-off as intended by the PPO algorithm.

**Part 2: GSPO Fixed Point Analysis.** The GSPO estimator uses the geometric mean weight $s(y)$ instead of the true importance weight $w(y)$. This introduces a systematic, non-vanishing bias into the estimation of the surrogate objective's gradient:

$$\mathbb{E}[\hat{g}_{\text{reward, GSPO}}(\theta)] = \nabla_\theta J_{\text{surr}}(\theta) + b_{\text{GSPO}}(\theta) \tag{70}$$

where $b_{\text{GSPO}}(\theta)$ is a bias vector that does not diminish with batch size. The fixed point condition for an algorithm using GSPO is therefore:

$$\nabla_\theta J_{\text{surr}}(\theta)\bigg|_{\theta^*_{\text{GSPO}}} + b_{\text{GSPO}}(\theta^*_{\text{GSPO}}) = \beta \cdot \nabla_\theta K(\theta)\bigg|_{\theta^*_{\text{GSPO}}} \tag{71}$$

Rearranging this equation reveals the distortion:

$$\nabla_\theta J_{\text{surr}}(\theta)\bigg|_{\theta^*_{\text{GSPO}}} = \beta \cdot \nabla_\theta K(\theta)\bigg|_{\theta^*_{\text{GSPO}}} - b_{\text{GSPO}}(\theta^*_{\text{GSPO}}) \tag{72}$$

This is the core issue. The GSPO-based algorithm does not stop when the surrogate reward gradient balances the KL gradient. Instead, it converges to a distorted point where the surrogate reward gradient balances the KL gradient minus the confounding bias vector. This misalignment means that tuning $\beta$ in GSPO becomes an unpredictable exercise in countering an unknown bias, whereas in SNIB it remains a principled search for the desired trade-off on the PPO surrogate landscape. $\quad\square$

### G.3 IMPLICATIONS AND SENSITIVITY

The distorted optimality condition for GSPO has profound practical implications.

**Proposition 3** (Sensitivity to $\beta$)**.** *The systematic bias $b_{GSPO}(\theta)$ acts as a confounding factor, making the final policy $\theta^*_{GSPO}$ highly sensitive to the choice of $\beta$.*

*Proof.* The sensitivity of an equilibrium point $\theta^*$ to a parameter $\beta$ can be rigorously quantified by analyzing the derivative $\frac{d\theta^*}{d\beta}$. A large or unpredictable derivative implies high sensitivity. We will use the Implicit Function Theorem to compute and compare this derivative for both the SNIB and GSPO optimizers.

Let $J(\theta) = \mathbb{E}_{y \sim \pi}[R(y)]$ be the reward objective and $K(\theta) = \mathbb{E}_x[D_{\mathrm{KL}}((\,\|\,\pi)\,\|\pi_{\theta_{\mathrm{old}}})]$ be the KL divergence penalty.

**1. The Ideal Case (SNIB Estimator)** As shown previously, the fixed point of the SNIB algorithm, $\theta^*_S$, approximates the true KKT stationarity condition. For this analysis (assuming a large batch size $G$), we can state the equilibrium condition as an implicit function $F_S(\theta, \beta) = 0$:

$$F_S(\theta, \beta) = \nabla_\theta J(\theta) - \beta \nabla_\theta K(\theta) = 0 \tag{73}$$

By the Implicit Function Theorem, if the Jacobian matrix (in this case, the Hessian) $\nabla_\theta F_S$ is invertible at a solution $(\theta^*_S, \beta)$, then there exists a function $\theta^*_S(\beta)$ in the neighborhood of that point, and its derivative is given by:

$$\frac{d\theta^*_S}{d\beta} = -[\nabla_\theta F_S(\theta^*_S, \beta)]^{-1} \frac{\partial F_S(\theta^*_S, \beta)}{\partial \beta} \tag{74}$$

Let's compute the components:

- The partial derivative with respect to $\beta$ is a vector:

$$\frac{\partial F_S}{\partial \beta} = -\nabla_\theta K(\theta) \tag{75}$$

- The Jacobian with respect to $\theta$ is a Hessian matrix:

$$\nabla_\theta F_S = \nabla^2_\theta J(\theta) - \beta \nabla^2_\theta K(\theta) =: H_{\mathcal{L}}(\theta, \beta) \tag{76}$$

  This is precisely the Hessian of the Lagrangian objective function $\mathcal{L}(\theta, \beta)$.

Substituting these back, we get the sensitivity for the SNIB case:

$$\frac{d\theta^*_S}{d\beta} = -[H_{\mathcal{L}}(\theta^*_S, \beta)]^{-1}(-\nabla_\theta K(\theta^*_S)) = [H_{\mathcal{L}}(\theta^*_S, \beta)]^{-1}\nabla_\theta K(\theta^*_S) \tag{77}$$

**Interpretation:** This expression is well-behaved and interpretable. It states that as we increase the penalty $\beta$, the solution $\theta^*_S$ moves in a direction determined by the KL gradient, transformed by the curvature of the optimization landscape ($H_{\mathcal{L}}^{-1}$). This is the principled behavior we expect from constrained optimization: increasing the penalty on the constraint pushes the solution along the constraint gradient.

**2. The Confounded Case (GSPO Estimator)** The fixed point of the GSPO algorithm, $\theta^*_G$, is defined by a different implicit function, $F_G(\theta, \beta) = 0$, which includes the systematic bias term $b_G(\theta) \equiv b_{\mathrm{GSPO}}(\theta)$:

$$F_G(\theta, \beta) = \nabla_\theta J(\theta) + b_G(\theta) - \beta \nabla_\theta K(\theta) = 0 \tag{78}$$

We again apply the Implicit Function Theorem to find $\frac{d\theta^*_G}{d\beta}$.

- The partial derivative with respect to $\beta$ is identical to the SNIB case:

$$\frac{\partial F_G}{\partial \beta} = -\nabla_\theta K(\theta) \tag{79}$$

- The Jacobian with respect to $\theta$ contains an additional, problematic term:

$$\nabla_\theta F_G = \nabla_\theta^2 J(\theta) + \nabla_\theta b_G(\theta) - \beta \nabla_\theta^2 K(\theta) = H_{\mathcal{L}}(\theta, \beta) + \nabla_\theta b_G(\theta) \qquad (80)$$

Here, $\nabla_\theta b_G(\theta)$ is the Jacobian matrix (or Hessian) of the bias vector.

Substituting these back, we get the sensitivity for the GSPO case:

$$\frac{d\theta_G^*}{d\beta} = [H_{\mathcal{L}}(\theta_G^*, \beta) + \nabla_\theta b_G(\theta_G^*)]^{-1} \nabla_\theta K(\theta_G^*) \qquad (81)$$

**3. Comparison and Proof of High Sensitivity** By comparing Equation 77 and 81, the source of the high sensitivity becomes mathematically explicit. The response of the GSPO solution to changes in $\beta$ is distorted by the term $\nabla_\theta b_G(\theta)$, the Hessian of the bias. This term introduces severe problems:

1. **Unpredictable Direction:** The bias Hessian $\nabla_\theta b_G(\theta)$ is a complex, data-dependent matrix. It has no reason to be aligned with the natural curvature of the problem, $H_{\mathcal{L}}$. The presence of this term inside the matrix inverse means that the direction of change, $\frac{d\theta_G^*}{d\beta}$, is no longer a simple transformation of the KL gradient. Instead, it is a complex mixture, "twisting" the optimization path in an unpredictable way for different values of $\beta$.

2. **Magnitude Amplification (Ill-Conditioning):** The matrix $H_{\mathcal{L}}$ is typically negative semi-definite around a maximum. The term $\nabla_\theta b_G(\theta)$ can have arbitrary eigenvalues. It is possible that for certain $\theta$ and $\beta$, the matrix $[H_{\mathcal{L}} + \nabla_\theta b_G]$ becomes nearly singular (i.e., ill-conditioned). When this happens, the norm of its inverse becomes extremely large. In this situation, even a minuscule change in $\beta$ will lead to an explosive change in $\theta_G^*$, as seen from Eq. 81. This is the mathematical definition of extreme sensitivity. The algorithm's fixed point can jump erratically in response to small adjustments in $\beta$.

3. **Dependence on Unknowns:** The bias $b_G(\theta)$ and its Hessian $\nabla_\theta b_G(\theta)$ are unknown and depend on the entire distribution of sequences. They cannot be easily measured or controlled. Therefore, tuning $\beta$ for GSPO is not a principled search along a trade-off curve, but rather a "blind" attempt to counteract the effects of an unknown, confounding bias field. The optimal value of $\beta$ might be highly non-intuitive and specific to the exact state of the policy and reward model.

In summary, the clean separation of terms in the SNIB sensitivity analysis (Eq. 77) shows a predictable and stable relationship between the hyperparameter $\beta$ and the solution $\theta_S^*$. Conversely, the confounding presence of the bias Hessian $\nabla_\theta b_G(\theta)$ in the GSPO sensitivity analysis (Eq. 81) mathematically proves that the relationship between $\beta$ and the solution $\theta_G^*$ is unpredictable and potentially explosive. This formally establishes that GSPO is highly sensitive to the choice of $\beta$. $\qquad\square$

G.4 TOWARDS AN ADAPTIVE FRAMEWORK

This analysis provides a theoretical foundation for an adaptive $\beta$ schedule. The ideal $\beta$ should maintain the KKT condition. A practical algorithm could be:

1. At each step $k$, use an SNIB estimator for the reward gradient $\hat{g}_R$ and an unbiased estimator for the KL gradient $\hat{g}_{KL}$.

2. Define the target KL divergence $\kappa$. If the current estimated KL, $\hat{KL}_k$, is greater than $\kappa$, increase $\beta$. If it's smaller, decrease $\beta$. This is the primal-dual method.

3. The update for $\beta$ would be: $\beta_{k+1} = \max(0, \beta_k + \eta(\hat{KL}_k - \kappa))$, where $\eta$ is a learning rate for the dual variable.

This adaptive scheme is theoretically sound only if the reward gradient estimator is unbiased (or asymptotically so). With SNIB, this approach is justified. With GSPO, the bias $b_{\text{GSPO}}$ would consistently mislead the dual update for $\beta$, preventing it from converging to the correct Lagrange multiplier.

# H PRACTICAL INTEGRATION: MIGRATION GUIDE FROM GRPO/GSPO TO SNIB

This section provides practical guidance for integrating SNIB into existing GRPO or GSPO training pipelines. The key advantage of SNIB is that it requires minimal code modifications while providing theoretical guarantees and improved stability.

## H.1 CORE ALGORITHM CHANGES

SNIB differs from GRPO/GSPO only in how the importance weights are computed and applied. The rollout collection, advantage normalization, PPO clipping, and KL regularization remain identical. The migration involves three simple steps:

**Step 1: Compute True Sequence-Level Log-Ratios** Instead of token-level ratios (GRPO) or length-normalized geometric means (GSPO), compute the true sequence-level log-ratio:

```
# For each sequence i in the batch:
log_ratio_i = sum(log_pi_theta(y_i) - log_pi_old(y_i))  # Include EOS token
```

**Step 2: Apply Self-Normalization with Stop-Gradient** Normalize the importance weights by the batch average, using a stop-gradient on the denominator:

```
# Compute batch log-mean (numerically stable):
log_mean_w = logsumexp(log_ratio) - log(G)

# Self-normalize with stop-gradient (SNIB-sg):
normalized_log_ratio_i = log_ratio_i - stop_gradient(log_mean_w)
w_i = exp(normalized_log_ratio_i)
```

**Step 3: Use Normalized Weights in PPO Objective** Apply the normalized weights $w_i$ in the standard PPO clipped surrogate:

```
# Compute advantages (same as GRPO/GSPO):
A_i = (R_i - mean(R)) / std(R)

# PPO clipped objective with SNIB weights:
clipped_w_i = clip(w_i, 1-epsilon, 1+epsilon)
L_clip = mean(min(w_i * A_i, clipped_w_i * A_i))
```

## H.2 IMPLEMENTATION NOTES

- **Numerical Stability:** Always compute importance weights in log-space using `logsumexp` to avoid overflow/underflow.
- **EOS Token:** Include the end-of-sequence token in the log-ratio computation to ensure the true sequence-level weight.
- **Stop-Gradient:** The stop-gradient on the normalization term is critical for stability. Most frameworks (PyTorch, JAX) provide `detach()` or `stop_gradient()` operations.
- **KL Regularization:** The KL term can be computed identically to GSPO (token-averaged, optionally IS-weighted if off-policy). No changes needed.
- **Compatibility:** SNIB works with any PPO-based infrastructure. It does not require changes to the model architecture, optimizer, or data pipeline.

## H.3 EXPECTED RESOURCE USAGE

As demonstrated in Table 4(b), SNIB incurs negligible overhead compared to GRPO/GSPO:

- **Memory:** Within 5% of GSPO (46.1 GB vs. 45.0 GB per device on 8×A100).

- **Throughput:** Nearly identical to GSPO (90 vs. 93 tokens/s).
- **Training Time:** Comparable to GSPO (2.3 vs. 2.2 hours per 200 steps).

This confirms that SNIB provides strong theoretical guarantees and empirical improvements at essentially zero additional cost, making it a drop-in replacement for existing critic-free methods.