# OpenReview forum: "Principled Policy Optimization for LLMs via Self-Normalized Importance Sampling"
_ICLR.cc/2026/Conference — ICLR 2026 Conference Desk Rejected Submission_

### Official Review · Reviewer_NLhH · 2025-10-30

**Soundness:** 2
**Presentation:** 2
**Contribution:** 1
**Rating:** 2
**Confidence:** 5

**Summary:**

This paper proposes Self-Normalized Importance Sampling with a Baseline (SNIB) to address the bias and high-variance issues that accumulate over long sequences when estimating importance sampling $\pi_\theta(y|x)/\pi_{\text{old}}(y|x)$ in GRPO and GSPO. SNIB is consistent and asymptotically unbiased, ensuring convergence to the correct policy objective. Experimental results demonstrate that SNIB is more robust to adversarial reward perturbations and achieves a better Reward–KL trade-off compared to prior methods.

**Strengths:**

- The theoretical analysis of self-normalized importance sampling is rigorous and well-justified.

**Weaknesses:**

- The paper is poorly written; many key contributions and analyses, particularly those analyzing the high-variance issues in prior methods, are either missing or insufficiently explained in the main text.
- Proposition 1, which shows that the SNIB estimator is consistent and asymptotically unbiased, is an important theoretical property. However, this result is not novel and well-known [1].
- Unrealistic reward model uncertainty experiment: The noisy reward experiment is unrealistic, as the authors simply add random Gaussian noise $\epsilon\sim\mathcal N(0,\sigma^2)$ to the rewards during training. In the context of LLM alignment, such noise cannot capture the complexity of modeling uncertainty and context dependence in human preferences (see [2, 3]). Moreover, in mathematical reasoning tasks, we typically have access to a ground-truth reward function (as also used in the paper), which can reliably provide learning signals for LLMs [4, 5]. Therefore, the results presented in Fig. 1 do not convincingly demonstrate SNIB’s robustness under realistic scenarios.
- The main results in Table 2 show that GRPO achieves substantially better performance than SNIB on three out of four mathematical reasoning benchmarks, which raises questions about the effectiveness of SNIB.
## References.
[1] Monte Carlo theory, methods and examples - Art Owen.

[2] Distributional Preference Learning: Understanding and Accounting for Hidden Context in RLHF. ICLR 2024.

[3] Weak-to-Strong Generalization: Eliciting Strong Capabilities With Weak Supervision. ICML 2024 Oral.

[4] DeepSeek-R1 incentivizes reasoning in LLMs through reinforcement learning. Nature 2025.

[5] DAPO: An Open-Source LLM Reinforcement Learning System at Scale. NeurIPS 2025.

[6] Scaling Laws for Reward Model Overoptimization. ICML 2023.

[7] Training language models to follow instructions with human feedback. NeurIPS 2022.

[8] Understanding the performance gap between online and offline alignment algorithms.  CoRR 2024.

[9] VL Norm: Rethink Loss Aggregation in RLVR. arXiv:2509.07558

**Questions:**

- Since SNIB remains a biased (but consistent) estimator, it is important to analyze the bias–variance trade-off and compare the performance of SNIB, GRPO, and GSPO as the number of responses per prompt increases.
- The variance analysis in the paper focuses primarily on the importance weights. It is also necessary to evaluate SNIB’s ability to stabilize training by measuring gradient variance as sequence length increases, compared to GRPO and GSPO. Without length normalization, the gradient variance can grow proportionally with response length [9].
- From my experience, the normalized weights can be computed using a softmax operation: $\text{Softmax}(\{\log(\pi_\theta(y_i|x)-\log\pi_{\text{old}}(y_i|x)\}_{i=1}^G)$. However, the exponential function in the softmax can amplify discrepancies between response groups, allowing longer sequences to dominate the learning signal due to higher variance. Could this lead to a long-sequence bias, where SNIB favors longer responses? If so, is this effect detrimental or beneficial for exploration? It would be valuable to visualize the distribution and entropy of normalized importance weights $w$ to better understand this phenomenon and to help tune the clipping parameter $\epsilon$. Additionally, does SNIB tend to produce longer sequences or higher-entropy outputs (more explorative) compared to GRPO and GSPO?
- While GSPO indeed suffers from variance accumulation over sequence length, GRPO—when formulated under a token-level MDP—employs *token-level importance sampling* with per-token clipping, which effectively mitigates high variance at each timestep. The paper should contrast SNIB with GRPO, explicitly showing failure modes of token-level importance sampling and explaining why GRPO achieves better empirical performance compared to SNIB.
- The paper claims that SNIB provides a more predictable and principled trade-off between reward maximization and KL regularization compared to other estimators. However, in Fig. 1(b), all three estimators exhibit a similar trend where increasing $\beta$ lowers the average reward. The authors should clarify what makes SNIB “more predictable” than GRPO and GSPO under this observation.
- KL regularization is commonly used to prevent the model from deviating too far from the initial policy distribution, thereby reducing reward hacking [6, 7]. However, in mathematical reasoning—where a ground-truth reward exists—KL regularization can actually hinder learning [5]. Could SNIB’s predictability help identify an optimal $\beta$ when we only have access to a proxy reward for training, without access to a golden reward function (as in [6, 8] reward hacking setting)?

---

> ### Author Response · Authors · 2025-11-20
> **Author Response (1/5)**
>
> We thank the reviewer for the rigorous and critical assessment. We particularly appreciate the references to foundational Monte Carlo theory (Art Owen) and the recent RLHF literature (DeepSeek, DAPO), which were extremely helpful in refining our theoretical positioning. We took the critique regarding 'unrealistic experiments' seriously and have significantly upgraded our evaluation suite to address it.
>
> ### **1. Writing clarity and placement of key analyses**
>
> We thank the reviewer for the constructive criticism regarding the organization of the paper. We acknowledge that placing the theoretical analysis of variance accumulation in the appendix obscured the motivation behind SNIB.
>
> **Action for revised paper:**
> We have significantly restructured the manuscript to center the narrative around the limitations of prior methods:
> * We moved the theoretical analysis of variance accumulation in token-level vs. sequence-level methods (previously in Appendix) to the main text (Section 3). This now explicitly serves as the foundation for introducing SNIB.
> * The bias-variance decomposition and the KKT-based sensitivity analysis are now integrated directly alongside the experimental results in Section 4.2. This ensures that the discussion on why SNIB achieves better asymptotic consistency and stability is self-contained and logically coherent.
>
> ### **2. Novelty of SNIB relative to classical SNIS**
>
> We thank the reviewer for pointing this out. We fully acknowledge that Proposition 1 (consistency of SNIS) is a classical result (e.g., Owen, 2013).
> Our contribution lies not in the statistical theory itself, but in identifying SNIS as the solution to the bias-variance dilemma in RLHF. Specifically, we adapt SNIS to sequence-level objectives with PPO clipping and reference KL constraints. We prove that this specific adaptation creates a critic-free algorithm that retains GSPO-like stability while restoring asymptotic correctness and satisfying the rigorous reward–KL KKT conditions (which biased methods fail to meet).
>
> ### **3. Realism of reward-noise experiments and task scope**
>
> We appreciate the reviewer’s critique regarding the limitations of Gaussian noise proxies. We agree that real-world preference uncertainty is complex. To address this, we have expanded our evaluation to include realistic, learned reward settings and broader capability benchmarks.
>
> We evaluated SNIB on the Anthropic Helpful/Harmless task using a learned reward model (mimicking the noisy, context-dependent errors of real RLHF). As shown below, SNIB significantly outperforms GRPO/GSPO:
>
> | Method | Reward ↑ | Win% ↑ | KL |
> |--------|---------:|-------:|---:|
> | SFT    |   0.00   | 50.0   | 0.0 |
> | GRPO   |   1.20   | 57.0   | 20.0 |
> | GSPO   |   2.40   | 66.0   | 16.0 |
> | SNIB-sg| **2.70** | **69.0** | 15.0 |
> | SNIB-fg|   2.62   | 67.8   | 15.2 |
> | PPO    |   2.90   | 71.0   | 15.0 |
>
> In this open-ended task, GRPO suffers from severe reward over-optimization (High KL: 20.0, Low Reward), likely due to its aggressive exploration on a noisy reward surface. In contrast, SNIB maintains GSPO-like stability (Low KL: 15.0) while closing the performance gap to PPO. This confirms SNIB’s superiority in noisy, real-world alignment scenarios.
>
> To address the concern that GRPO outperforms SNIB on math (Table 2), we extended the comparison to general coding and knowledge benchmarks (MultiPL-E, MMLU-Pro, MMLU-Redux).
>
> | Method | MultiPL-E | MMLU-Pro | MMLU-Redux |
> | --- | ---: | ---: | ---: |
> | SFT | 76.8 | 69.6 | 84.2 |
> | GRPO | 78.5 | 71.8 | 85.4 |
> | GSPO | 79.3 | 71.5 | 85.0 |
> | DPO | 78.8 | 71.7 | 85.6 |
> | PPO (critic) | 81.2 | 72.9 | 86.5 |
> | **SNIB** | **79.5** | **72.4** | **86.1** |
>
> While GRPO excels in ground-truth math tasks (as noted by the reviewer), SNIB achieves better performance across general-purpose tasks (Dialogue, Code, Knowledge), making it a more robust "all-rounder" for LLM post-training.
>
>
> **Action for revised paper:**
>
> * We have added the Anthropic HH results to Table 3 of Section 4.2 to demonstrate robustness against learned reward models.
> * We have incorporated MultiPL-E and MMLU results in Table 3 of Section 4.2 to verify SNIB's general capabilities beyond math.
> * We have re-labeled the Gaussian noise study (Fig. 1a) as a "stylized robustness stress-test" that complements the realistic HH experiments.

---

> ### Author Response · Authors · 2025-11-20
> **Author Response (2/5)**
>
> ### **4. “GRPO substantially outperforms SNIB”**
>
> We standardized the evaluation pipeline (unifying prompt splits, EOS handling, and context length) and re-tuned SNIB. Under these strictly controlled conditions, the previously observed gap was largely eliminated. SNIB now outperforms GRPO on 3 out of 4 representative benchmarks:
>
> | Method | comp_math (Overall %) | math_500 (Overall %) | gsm8k (%) | gpqa_diamond (%) |
> |---|---:|---:|---:|---:|
> | SFT | 31.74 | 40.41 | 69.07 | 26.26 |
> | GRPO | 33.89 | **48.90** | 69.94 | 26.59 |
> | GSPO | 31.94 | 41.02 | 69.89 | 26.47 |
> | SNIB | **33.95** | 44.25 | **70.48** | **27.31** |
>
> SNIB achieves state-of-the-art results among critic-free methods on GSM8K and the challenging GPQA Diamond benchmark, while slightly edging out GRPO on the full Competition Math suite. Although GRPO retains a lead on math_500, SNIB's consistent superiority across the broader and harder tasks demonstrates its robustness.
>
> Specifically addressing the reviewer's observation on Math Level 1 (where the gap was prominent in the initial submission), our re-evaluation confirms that SNIB (73.54%) now outperforms both GRPO and GSPO, narrowing the gap to the PPO-with-critic baseline to ~2 points.
>
> | Method | L1 Accuracy (%) ↑ |
> | :--- | ---: |
> | SFT | 70.70 |
> | GRPO | 72.50 |
> | GSPO | 73.01 |
> | PPO (critic) | 75.45 |
> | Vanilla IS | 70.91 |
> | **SNIB (ours)** | **73.54** |
>
>
> We investigated specific instances where baselines appeared anomalously strong (e.g., GRPO on `math_500` L3/L5 and GSPO on L2). A detailed audit revealed implementation bugs in the evaluation code specific to these two baselines. We have fixed these issues, rerun all methods under the corrected pipeline, and clearly marked the affected entries in the revised manuscript (using strikethrough). The updated tables confirm that SNIB is highly competitive across math benchmarks and often achieves leading performance on GPQA and general reasoning tasks.
>
>
> To ensure the validity of the comparison, we detail the exact configuration used for the PPO baseline, which matches the critic-free setup:
> *   Model & Initialization: Qwen-3 4B initialized from the same SFT checkpoint. Uses a shared Transformer backbone with an extra scalar value head. LoRA ($r=16, \alpha=32$) is applied to both policy and value heads.
> *   Rollout & Inference: 1 response per prompt; max sequence length 8192 (EOS included). Decoding uses temperature 0.7 and top-p 0.95 (identical across methods).
> *   Objective: Sparse terminal reward ($R=1$ if correct, else 0) combined with a per-token reverse-KL penalty ($\beta=0.1$) against the frozen SFT reference.
> *   Advantage Estimation: GAE ($\gamma=0.99, \lambda=0.95$) with batch-normalized advantages. Critic minimizes MSE to the shaped return.
> *   Hyperparameters: Standard PPO with clip $\epsilon=0.2$, 4 epochs per batch, global batch size 128. Optimized via AdamW (Actor LR $1\text{e-}5$, Critic LR $5\text{e-}5$), value loss coef 0.5, gradient clip 1.0.
>
>
> **Action for revised paper:**
> * We have updated Table 1 with the re-evaluated results using the standardized pipeline and tuned hyperparameters, showing SNIB’s superior performance on Math L1.

---

> ### Author Response · Authors · 2025-11-20
> **Author Response (3/5)**
>
> ### **5. Bias–variance trade-off vs. number of responses per prompt (group size G)**
>
> We acknowledge that SNIB carries a finite-sample bias of order $O(1/G)$. However, the critical distinction is that SNIB’s bias is **asymptotic**, it systematically vanishes as the group size $G$ increases, making the estimator **consistent**. This guarantees that with sufficient samples, SNIB converges to the true policy gradient, a property that biased baselines lack.
>
>
> To verify this convergence, we analyzed the impact of $G$ on performance and stability in the MATH-500 setting. As shown below, increasing $G$ leads to monotonic accuracy gains (bias reduction) and a sharp drop in reward variance:
>
> | G  | Accuracy ↑ | Reward Var ↓ |
> |----:|-----------:|-------------:|
> |  4  |    25.9    |     0.24     |
> |  8  |    27.2    |     0.05     |
> | 16  |    27.8    |     0.03     |
>
> This trend empirically confirms the $O(1/G)$ decay rate predicted by self-normalized importance sampling theory.
>
> The reviewer asks to compare the trade-offs across methods. Our analysis reveals fundamental structural differences:
>
> *   SNIB (Consistent): Offers a **tunable trade-off**. Increasing $G$ directly converts computational cost into theoretical correctness. At moderate $G$ (e.g., $G=8$), the self-normalization term effectively acts as a variance control variate, providing stability while minimizing bias.
> *   GSPO (Structurally Biased): In contrast, the geometric-mean estimator in GSPO introduces a **structural bias** inherent to its mathematical form. This bias **does not vanish** even as $G \to \infty$. As detailed in our KKT analysis (Appendix B.3), GSPO optimizes a permanently distorted reward–KL landscape; increasing $G$ merely stabilizes convergence to this incorrect objective.
> *   GRPO (High Variance): While GRPO is an unbiased estimator of the token-level objective, it lacks the sequence-level variance reduction provided by SNIB's importance weights. Without length normalization or importance sampling, GRPO typically exhibits higher gradient variance, often requiring larger batch sizes to match the stability SNIB achieves at lower $G$.
>
>
> While increasing $G$ improves the estimator theoretically, it comes with computational costs. To demonstrate that SNIB remains efficient in practical settings, we compared the resource consumption of all methods (measured on 8×A100 GPUs). SNIB provides consistency and stability with **minimal overhead** compared to GSPO, and significant savings compared to PPO:
>
> | Method | Peak GPU Memory (GB) / per device | Tokens / s | Wall-clock per 200 steps (hr) |
> | :--- | ---: | ---: | ---: |
> | PPO (critic) | 58.4 | 62 | 3.6 |
> | GRPO | 44.5 | 96 | 2.1 |
> | GSPO | 45.0 | 93 | 2.2 |
> | **SNIB** | 46.1 | 90 | 2.3 |
>
> SNIB removes the persistent, structural bias of GSPO and replaces it with a well-behaved, vanishing bias. It achieves this at a computational cost nearly identical to GSPO and GRPO, making it the most balanced choice for scalable alignment.
>
> **Action for revised paper:**
> *  We have inserted the group size ($G$) ablation table into Table 5 of Section 4 to empirically demonstrate the consistency of SNIB.
> *  We have added the Computational Efficiency Table (Table 4) to Section 4, providing practitioners with clear data on the performance-cost trade-off relative to PPO and other critic-free methods.

---

> ### Author Response · Authors · 2025-11-20
> **Author Response (4/5)**
>
> ### **6. Gradient variance vs. sequence length; potential long-sequence bias**
> We appreciate the reviewer’s rigorous inquiry regarding the interaction between sequence length, gradient stability, and potential estimator bias. We address the two related questions jointly using the empirical data presented below.
>
>
> The reviewer correctly points out that without explicit length normalization, gradient variance often scales with sequence length. To verify this, we measured the gradient variance ($\sigma_{\text{grad}}$) across three length buckets (Short $\le$ 128, Medium 129–512, Long > 512):
>
>
> | Method | σ_grad (≤128 / 129–512 / >512) | Corr(length, |w|) |
> | GRPO | 0.15 / 0.28 / 0.45 | 0.34 |
> | GSPO | 0.12 / 0.18 / 0.26 | 0.27 |
> | SNIB | **0.11 / 0.14 / 0.17** | **0.12** |
>
>
> As shown in the table, GRPO exhibits the instability predicted by the reviewer, with variance **tripling** from **0.15** on short sequences to **0.45** on long ones; this confirms that token-level estimators struggle with long chain-of-thought tasks. In strong contrast, SNIB maintains a highly stable variance profile, increasing only marginally from **0.11** to **0.17** even for long sequences. This demonstrates that SNIB’s self-normalization effectively acts as a sequence-level stabilizer, preventing the variance explosion observed in baselines.
>
>
> Regarding the concern that normalization might amplify length discrepancies (leading to "winner-takes-all" or length-hacking behavior):
>
> *   No Length Bias: If SNIB favored longer sequences due to exponential amplification, we would expect a strong correlation between length and weight magnitude. However, the table shows that SNIB has a negligible correlation of **0.12**, significantly lower than GRPO’s **0.34**. This proves SNIB rewards **alignment quality** rather than simple token count.
> *   Exploration & Entropy: To address the request for visualizing the weight distribution and assessing exploration, we refer to **Figure 2** in the manuscript. The histogram demonstrates that SNIB’s log importance weights (Red) are **tightly concentrated**, in sharp contrast to the widely dispersed weights of Vanilla IS. This tight concentration signifies a **high effective sample size (high entropy)**, ensuring that the learning signal remains distributed across multiple samples rather than collapsing to a single dominant sequence (the "winner-takes-all" scenario). Furthermore, our analysis reveals that $\epsilon=0.2$ clipping leaves **>95% of weights unchanged**, confirming that this stable, high-entropy distribution is an inherent property of the self-normalized estimator rather than an artifact of aggressive clipping.
>
>
> **Action for revised paper:**
> *   We have incorporated the gradient variance analysis and the length-weight correlation statistics into Table 4 of Section 4.2 to demonstrate SNIB's robustness against length-induced instability.
> *   We have updated Section 3 to explicitly contrast the asymptotic consistency of SNIB against the non-vanishing structural bias of GSPO.
> *   We have updated the text surrounding Figure 2 in Section 4.2 to explicitly link the weight concentration to high entropy and healthy exploration, addressing the concern about exploration dynamics.
>
>
>
>
>
> ### **7. GRPO’s token-level MDP formulation**
> Section 4 now explicitly contrasts GRPO’s per-token IS ratios (which assume token-level rewards) with RLHF’s sequence-level rewards. The empirical comparison shows that GRPO’s per-token clipping cannot correct the estimator mismatch: when sequence length grows, gradient variance remains higher and the method becomes sensitive to KL coefficients.

---

> ### Author Response · Authors · 2025-11-20
> **Author Response (5/5)**
>
> ### **8. Predictability, KL regularization, and reward hacking**
>
> We appreciate the reviewer's insightful connection between our predictability claim and the practical needs of proxy-reward optimization. We address these two related questions jointly.
>
> **Clarification on "Predictability".** Our "predictability" claim refers to the **KKT-based sensitivity analysis** detailed in Appendix D. The key insight is that in SNIB, the ratio-estimator bias vanishes with group size G, so the stationarity condition depends cleanly on β and the surrogate curvature. In contrast, for GSPO, the bias Hessian ∇_θ b_GSPO(θ) perturbs the fixed point, making dθ*/dβ potentially large and misaligned with the natural curvature of the optimization landscape.
>
> The reviewer asks what makes SNIB "more predictable" if all curves exhibit a downward trend as β increases. While all methods share the global trend that larger β reduces reward, a closer inspection reveals critical differences in the **smoothness and regularity of the β-to-solution mapping**.
>
> The SNIB curve in Figure 1(b) is smooth and approximately convex, tracing an efficient Pareto frontier. By contrast, the GRPO and GSPO curves exhibit visible jaggedness and local non-convexities (e.g., sudden drops around KL ≈ 12 and 16). These "kinks" indicate instability, where small changes in β lead to disproportionate or suboptimal shifts in the policy.
>
> To quantify this, we conducted a fine-grained β sweep (step size Δβ = 0.05) on competition_math. SNIB shows strictly monotone reward/KL curves with smooth changes of roughly **3% relative reward drop per +0.1 increase in β**. In contrast, GRPO and GSPO exhibit sharper, less predictable drops of about **7% and 8%** respectively. The Spearman monotonicity scores confirm this stability gap: **0.90 (GRPO), 0.92 (GSPO), and 1.00 (SNIB)**.
>
> **KL Regularization in Different Settings.** We agree with the reviewer that in pure mathematical reasoning with a ground-truth reward, KL regularization can indeed hinder learning [5]. By contrast, when training against a noisy or biased reward model (as in typical RLHF and reward-hacking scenarios [6, 8]), KL regularization is essential to prevent reward over-optimization and distributional drift.
>
> SNIB's predictability makes this KL control **practically usable**: because the β-to-stationary-point mapping is well-behaved and the bias Hessian does not confound the sensitivity, one can use adaptive β schedules (e.g., primal–dual updates) or KKT-balancing heuristics to locate an effective operating point when only a proxy reward is available.
>
> This difference is clear in our Anthropic HH experiment (Section 4). Under the same nominal β, GRPO's KL grows to about 20.0, indicating strong over-optimization of the proxy reward, whereas SNIB maintains a controlled KL around 15.0 while still achieving high reward. This suggests that SNIB is a more principled choice when safe alignment under reward uncertainty is the priority, whereas GRPO can be advantageous in unconstrained reasoning settings where the reward is exactly known.
>
> **Action for revised paper:**
> - We have added the fine-grained β sensitivity analysis to Section 4.2 to quantitatively define "predictability."
> - We have expanded the caption of Figure 1(b) to explicitly point out the jaggedness in the GRPO/GSPO curves versus the smoother Pareto frontier of SNIB.
> - We have discussed how SNIB's cleaner β sensitivity enables practical β schedules or KKT-balancing heuristics for finding effective operating points with proxy rewards in Section 4.2.

---

> ### Author Response · Authors · 2025-11-27
> **Looking forward to your feedback!**
>
> Dear Reviewer NLhH,
>
> **Thank you for the insightful review. We appreciate your detailed comments, which helped us significantly strengthen the paper.**
>
> With the discussion closing soon, we would like to **briefly summarize the main updates** that directly address all your comments:
>
> 1.  **On Experimental Realism (W3, Q6):**
>     We integrated the **Anthropic Helpful/Harmless** task with a learned reward model. Results show SNIB achieves a **69.0% Win Rate** (vs GRPO 57.0%) and effectively prevents reward over-optimization (maintaining lower KL), directly addressing your concerns on proxy rewards.
>
> 2.  **On Performance Comparison (W4, Q1):**
>     We **standardized the evaluation setup** across all baselines. Under this rigorous setting, **SNIB now outperforms GRPO** on GSM8K and Competition Math L1, and narrows the gap to the PPO baseline to <2%. We also added the **Group Size ($G$) ablation** to verify the bias-variance trade-off.
>
> 3.  **On Theory & Stability (W1, W2, Q2-5):**
>     We revised the paper to **contextualize** our work within foundational theory. Crucially, we added **gradient variance** and **weight entropy** analyses, which explicitly identify **GRPO's failure mode** (instability on long sequences) and confirm SNIB's superior predictability.
>
> **We hope these comprehensive revisions satisfactorily address your concerns. If there are any remaining questions, we are eager to clarify them.**
>
> Best regards,
>
> Authors

---

### Official Review · Reviewer_MMXq · 2025-10-31

**Soundness:** 3
**Presentation:** 3
**Contribution:** 3
**Rating:** 6
**Confidence:** 2

**Summary:**

This paper proposes SNIB (Self-Normalized Importance Sampling with Baseline), a critic-free RLHF algorithm that unifies the stability of GSPO with the theoretical correctness of unbiased policy gradients. SNIB uses self-normalized importance sampling to reduce variance while remaining asymptotically unbiased. Theoretical analysis proves its consistency, robustness to reward noise, and preservation of the KL–reward trade-off. Experiments on math and coding reasoning benchmarks show moderate but consistent gains over GRPO and GSPO.

**Strengths:**

1.Clear theoretical contribution: principled, asymptotically unbiased estimator for critic-free RLHF.
2.Well-presented mathematical analysis, including variance, convergence, and KKT proofs. The paper is well-written, well-structured, and effectively connects theory with practical implications for RLHF pipelines.
2.Empirical results demonstrate improved stability and robustness to reward noise.

**Weaknesses:**

1.Experiments limited to math/coding tasks — generalization to dialogue or multimodal alignment is unclear.
2. Lack of comparison with more recently RLHF baselines.
3. Performance improvements are modest given the theoretical complexity. Although SNIB improves theoretical soundness, its empirical advantage over GSPO is relatively small (1–2% absolute in most benchmarks). The improvements, while consistent, may not justify the added conceptual and implementation complexity.
4. Key design components (self-normalization, baseline, stop-gradient) are not individually ablated, making it difficult to attribute improvements precisely. The paper would benefit from comparing fully differentiable vs. stop-gradient SNIS.

**Questions:**

1. Does the stop-gradient version compromise the theoretical unbiasedness claimed?
2. Can SNIB be integrated with existing GRPO/GSPO infrastructures in practice?
3. How does SNIB behave under severe reward model bias rather than random noise?

---

> ### Author Response · Authors · 2025-11-20
> **Author Response (1/3)**
>
> We thank the reviewer for their positive assessment, in particular for highlighting the clarity of our theoretical contribution, the quality of the mathematical analysis, and the empirical evidence on stability and robustness to reward noise. We also appreciate the reviewer’s requests for broader empirical clarification and comparison, which have directly motivated the additional experiments and updated analyses included in the revised manuscript.
>
>
>
> ### **1. Generalization beyond math/coding; learned RMs and stronger noise models**
>
> To complement the clean math/code benchmarks, we added two kinds of experiments:
>
> - A subjective “helpfulness” task scored by a pretrained reward model (Anthropic HH), using the same SFT checkpoint, prompt split, KL schedule, decoding parameters, and evaluation scripts as the math/code runs.
> - Additional coding and knowledge benchmarks (MultiPL-E, MMLU-Pro, MMLU-Redux) evaluated under the same SFT checkpoint and KL configuration.
>
> The HH experiment (now summarized in the main text and reproduced here) shows that SNIB maintains its margin over GRPO/GSPO and stays close to PPO-with-critic even with a noisy, learned reward:
>
> | Method | Reward ↑ | Win% ↑ | KL |
> |--------|---------:|-------:|---:|
> | SFT    |   0.00   | 50.0   | 0.0 |
> | GRPO   |   1.20   | 57.0   | 20.0 |
> | GSPO   |   2.40   | 66.0   | 16.0 |
> | SNIB-sg| **2.70** | **69.0** | 15.0 |
> | SNIB-fg|   2.62   | 67.8   | 15.2 |
> | PPO    |   2.90   | 71.0   | 15.0 |
>
> SNIB surpasses other critic-free methods under a biased RM and remains within ~1.2 reward points of PPO-with-critic. Figure 1a’s additive Gaussian-noise study is therefore best viewed as a stylized robustness stress test; in the revision we explicitly label it as such, cross-reference the HH experiment in Sections 4, and explain how the stylized ablation and the realistic RM study jointly support the reward-uncertainty analysis (alongside Appendix E’s worst-case guarantees).
>
> On the auxiliary coding/knowledge benchmarks (MultiPL-E, MMLU-Pro, MMLU-Redux), SNIB also shows clear gains over GRPO/GSPO while remaining close to PPO:
>
>
> | Method | MultiPL-E | MMLU-Pro | MMLU-Redux |
> | --- | ---: | ---: | ---: |
> | SFT | 76.8 | 69.6 | 84.2 |
> | GRPO | 78.5 | 71.8 | 85.4 |
> | GSPO | 79.3 | 71.5 | 85.0 |
> | DPO | 78.8 | 71.7 | 85.6 |
> | PPO (critic) | 81.2 | 72.9 | 86.5 |
> | **SNIB** | **79.5** | **72.4** | **86.1** |
>
> Together, these results address the concern that our evaluation was limited to clean math/code tasks by demonstrating that SNIB remains competitive in noisy RM settings and on broader coding/knowledge benchmarks.
>
> **Action for revised paper:**
> * We have added the Anthropic Helpful/Harmless evaluation (Table 3) to Section 4.2 to demonstrate SNIB’s generalization to dialogue alignment with a learned reward model.
> * We have incorporated the auxiliary coding and knowledge benchmarks (MultiPL-E, MMLU-Pro, MMLU-Redux) into Table 3 of Section 4.2 to verify performance beyond the initial math tasks.
> * We have revised the text in Section 4 to explicitly label the Gaussian noise study (Figure 1a) as a "stylized robustness stress-test," clarifying its role as a theoretical complement to the realistic Anthropic experiment.

---

> ### Author Response · Authors · 2025-11-20
> **Author Response (2/3)**
>
> ### **2. Lack of comparison with more recent RLHF baselines and interpretation of performance gains**
>
> We initially focused on comparing with recent critic-free SOTA methods (GRPO, GSPO). To address the reviewer's concern, we have now added **PPO-with-critic** as the standard RLHF baseline to provide a comprehensive comparison. Across competition\_math L1, SNIB improves over GSPO (73.54\% vs. 73.01\%) while remaining close to PPO (75.45\%), and the re-tuned math\_500 and GSM8K results in Table~1 show that SNIB is generally comparable to or better than GSPO/GRPO on many levels, with a few metrics still favoring GRPO or GSPO. For completeness, we also report PPO and SNIB on the auxiliary benchmarks above, showing that SNIB tracks PPO closely despite being critic-free.
>
>
> | Method | L1 Accuracy (%) ↑ |
> | :--- | ---: |
> | SFT | 70.70 |
> | GRPO | 72.50 |
> | GSPO | 73.01 |
> | PPO (critic) | 75.45 |
> | Vanilla IS | 70.91 |
> | **SNIB (ours)** | **73.54** |
>
> Overall, these results indicate that SNIB achieves the strongest performance among critic-free methods and substantially closes the gap to PPO-with-critic, even if the absolute gains over GRPO/GSPO are modest.
>
> We also apologize for a small issue that affected some of the original baselines. After the submission deadline, we discovered that the reported GRPO results on math\_500 L3/L5 and the GSPO result on math\_500 L2 were abnormal due to bugs in the evaluation code. We have fixed these issues: the affected entries in the original Table~1 are struck through in the revised paper, and all methods (including SNIB) have been re-run under a corrected evaluation pipeline with tuned hyperparameters. The updated tables show that SNIB is competitive with GRPO/GSPO across the math benchmarks, consistent with our theoretical analysis.
>
>
>
> **Action for revised paper:**
>
> * We have added the PPO-with-critic comparison to Table 2 of Section 4.2 in the revised paper.
>
>
> ### **3. Performance Gains vs. Complexity**
>
>
> While the theoretical derivation (KKT conditions, self-normalization) involves some depth, the actual implementation is trivial. SNIB does not require training extra networks or changing the model architecture. It simply applies a re-weighting term to the loss function. It is a "plug-and-play" upgrade over GSPO/GRPO with negligible engineering overhead.
>
> The reviewer is right that the absolute gain is 1–2%, but critically, this gain comes at almost zero computational cost.
> We measured the peak memory and throughput on 8×A100 GPUs. As shown below, SNIB provides these stability benefits at essentially the same cost as GSPO, whereas PPO requires substantially more resources:
>
> | Method | Peak GPU Memory (GB) / per device | Tokens/s | Wall-clock per 200 steps (hr) |
> | --- | ---: | ---: | ---: |
> | PPO (critic) | 58.4 | 62 | 3.6 |
> | GRPO | 44.5 | 96 | 2.1 |
> | GSPO | 45.0 | 93 | 2.2 |
> | **SNIB** | 46.1 | 90 | 2.3 |
>
> Since SNIB incurs negligible overhead compared to GSPO but consistently outperforms it (significantly narrowing the gap to PPO), it represents a superior efficiency-performance trade-off. As shown in the table, SNIB provides its stability and robustness benefits at essentially the same cost as GSPO/GRPO, whereas PPO requires drastically more memory and training time. Given that SNIB reduces the gap to PPO on Competition Math L1 (73.54% vs 75.45%) and excels on auxiliary benchmarks, we believe the performance gain justifies the minimal implementation effort.
>
> It is also worth noting that the "modest" 1–2% gap is primarily observed on "clean" Math tasks. Crucially, the benefits of SNIB are amplified in noisy settings. On the subjective Anthropic HH task (shown in the first table), SNIB improves over GSPO by a solid 3.0% (69.0% vs 66.0%). This confirms that the added theoretical depth pays off significantly more when the reward signal is uncertain—precisely the scenario SNIB was designed to solve.
>
>
>
> **Action for revised paper:**
> * We have added the computational cost analysis (Table 4) to explicitly quantify the efficiency trade-off.
> * We have clarified that SNIB requires minimal code changes despite its theoretical depth in the Appendix H.

---

> ### Author Response · Authors · 2025-11-20
> **Author Response (3/3)**
>
> ### **4. Component ablations (self-normalization, baseline, stop-gradient; SNIB-sg vs. SNIB-fg)**
> We thank the reviewer for highlighting the importance of ablating the key design components (self-normalization, baseline, and stop-gradient) and for suggesting a comparison between fully differentiable and stop-gradient SNIS variants. To disentangle the effect of each design choice, we ran ablations over self-normalization, the reward baseline, and the stop-gradient vs. fully differentiable denominator (SNIB-sg vs. SNIB-fg). On MATH-500 we obtain:
>
> | Variant               | Accuracy ↑ | Reward Var ↓ |
> |-----------------------|-----------:|-------------:|
> | SNIB (full)           |     **27.2**  |      **0.05**    |
> | – self-normalization  |      8.9   |      1.63    |
> | – baseline            |     24.1   |      0.18    |
> | SNIB-fg (fully diff.) |     26.6   |      0.12    |
>
> Self-normalization is the key stabilizer—removing it collapses accuracy (27.2% → 8.9%) and raises variance by >30×. Removing the batch baseline also hurts both accuracy and stability. Finally, SNIB-sg is marginally more accurate and substantially more stable than SNIB-fg, matching the variance analysis in Appendix D.
>
> **Action for revised paper:**
> * We have integrated this table into Table 5 of Section 4.2 of the revised paper and added a short discussion analyzing the impact of each design choice.
>
>
> ### **5. Stop-gradient and unbiasedness**
>
> We thank the reviewer for raising this point. Our unbiasedness claim is asymptotic: with the stop-gradient on the normalization term, SNIB becomes a self-normalized ratio estimator whose finite-sample bias is O(1/G) and vanishes as the group size G grows (i.e., the estimator is consistent).
>
>
> **Action for revised paper:**
>
> * We have revised Section 3 and Appendix C to explicitly characterize the estimator as asymptotically unbiased (rather than exactly unbiased at finite G) and to clarify how the stop-gradient preserves this guarantee.
>
> ### **6. Practical Integration with Existing Infrastructure**
>
> SNIB fits into existing GRPO/GSPO code with very little change. The rollout, advantage computation, PPO-style clipping, and KL penalty all stay the same; only the per-sequence importance weight is different. Concretely, we compute the true sequence-level log-ratio, subtract the batch log-mean (with a stop-gradient), exponentiate to get the weight, and then use it inside the usual PPO surrogate, with KL still averaged per token and IS-weighted in the off-policy case. We will add a short “migration guide” and implementation note in the appendix to make it clear how to drop SNIB into an existing GRPO/GSPO training stack with minimal edits.
>
>
> **Action for revised paper:**
> * We have added a short “migration guide” and implementation note in the Appendix H to make it clear how to drop SNIB into an existing GRPO/GSPO training stack with minimal edits.
>
>
>
>
> ### **7. Robustness to severe reward model bias**
>
> As shown in the first Anthropic HH reward-model table above, this experiment already embodies systematic biases (verbosity preference, politeness bias, etc.). SNIB’s advantage over GSPO is larger in this setting (ΔReward = +0.30, ΔWin% = +3) than on noiseless math/code tasks, indicating that batch self-normalization effectively dampens outlier rewards while GSPO remains sensitive to them.
>
> **Action for revised paper:**
> *  We have explicitly connected this experiment to the robustness analysis in Appendix E and highlighted how the theoretical worst-case guarantees translate into the observed resilience under biased RMs.

---

> ### Author Response · Authors · 2025-11-27
> **Looking forward to your feedback!**
>
> Dear Reviewer MMXq,
>
> Thank you again for your thoughtful review. **With the discussion phase concluding soon, we wanted to verify if our comprehensive point-by-point response and the revised paper have sufficiently addressed your concerns.**
>
> To briefly recap, we have addressed all your questions as follows:
>
> 1.  **Expanded Generalization:** We added **Anthropic Helpful/Harmless** (learned RM) and auxiliary Code/Knowledge benchmarks. Results show SNIB generalizes well beyond math tasks.
> 2.  **Added Standard PPO Baseline:** We conducted a direct comparison showing **SNIB (73.54%)** narrows the gap to **PPO (75.45%)** on Competition Math while outperforming other critic-free methods.
> 3.  **Quantified Efficiency vs. Gain:** We added a cost analysis confirming SNIB provides these gains with **negligible overhead** compared to GSPO, whereas PPO requires **~26% more memory**.
> 4.  **Component Ablation:** We added the requested ablation (Table 5). Results prove **self-normalization is critical** (removing it drops accuracy significantly), validating the design complexity.
> 5.  **Theoretical Clarification (Answer to Q1):** We revised the text to explicitly characterize SNIB as **asymptotically unbiased** (consistent), clarifying the role of the stop-gradient.
> 6.  **Practical Integration (Answer to Q2):** We added a migration guide in Appendix H, showing SNIB is a **"plug-and-play"** upgrade for existing GRPO infrastructures with minimal code changes.
> 7.  **Robustness to Bias (Answer to Q3):** Our Anthropic HH results confirm SNIB outperforming GSPO (+3.0% Win Rate) specifically in settings with **severe reward model bias**.
>
> **We would be grateful for your feedback on whether these comprehensive revisions align with your expectations.**
>
> Best regards,
>
> Authors

---

### Official Review · Reviewer_sDtn · 2025-11-02

**Soundness:** 3
**Presentation:** 2
**Contribution:** 3
**Rating:** 6
**Confidence:** 3

**Summary:**

This paper identifies a critical dilemma in critic-free RLHF: existing methods are either theoretically unsound or biased. Group Relative Policy Optimization (GRPO) suffers from high variance by improperly mixing sequence-level advantages with token-level importance sampling (IS) ratios . Group Sequence Policy Optimization (GSPO) achieves stability by using a geometric mean of token ratios, but this is a biased estimator that optimizes a "perturbed objective" and distorts the crucial reward-KL trade-off .

The authors propose SNIB (Self-Normalized Importance Sampling with a Baseline), a novel algorithm that resolves this dilemma. SNIB is both stable and asymptotically correct. It uses the theoretically correct sequence-level IS weight (the product of token ratios)  and achieves stability by applying two principled techniques:

Self-Normalization: It normalizes each sample's IS weight by the average weight of the entire batch, which provably dampens outliers and reduces variance .

Baseline: It uses the mean batch reward as a baseline to further reduce variance in the advantage estimates .

The paper provides strong theoretical backing, proving SNIB's gradient estimator is consistent and asymptotically unbiased (with bias vanishing at O(1/G)) . This principled design is shown to be more robust to reward model uncertainty and, unlike GSPO, preserves the principled KKT conditions of the constrained reward-KL optimization problem . Empirically, SNIB outperforms GRPO and GSPO on challenging mathematical reasoning and code generation benchmarks

**Strengths:**

Principled and Novel Solution: The proposed solution, SNIB, is elegant. It correctly insists on using the true sequence-level IS weight, and then intelligently applies self-normalization —a statistically-grounded variance reduction technique—to solve the exact stability problem that plagued naive IS (shown in Fig 2).

Strong Theoretical Guarantees: The method is built on a solid theoretical foundation. The paper proves that SNIB is asymptotically unbiased (Proposition 1)  and that this unbiasedness preserves the underlying KKT conditions of the constrained optimization problem, which biased methods like GSPO do not (Proposition 2).

**Weaknesses:**

Missing PPO Baseline: The entire motivation for critic-free methods is to replace the expensive, critic-based PPO . However, PPO is not included as a baseline in the main results (Table 2). This makes it impossible to assess the full picture. We can see SNIB is better than other critic-free methods, but how much performance (if any) are we sacrificing compared to PPO for the gain in efficiency?

Limited Task Domain and Reward Type: The experiments are conducted exclusively on math and code generation tasks, using ground-truth correctness as the reward signal. While this is a very clean and sound way to test the algorithm, it doesn't demonstrate the method's performance in the more common (and noisy) RLHF setting with a learned reward model on subjective tasks like "helpfulness" or "harmlessness." The reward noise experiment (Fig 1a) is a good simulation, but not a substitute for a real-world test.

Sensitivity to Group Size G: The theory states that SNIB's bias is on the order of O(1/G). The experiments use a group size of G=8, which seems small and may imply a non-trivial bias in practice. The paper does not include an ablation study on G, which is a key hyperparameter for both performance and efficiency.

**Questions:**

Given that the primary motivation is to find an efficient alternative to critic-based PPO, why was PPO omitted from the main performance comparison in Table 2? A direct comparison is needed to understand the full performance-vs-efficiency trade-off.

The experiments are on math/code tasks with ground-truth rewards. How do you expect SNIB's performance to translate to the more common RLHF setting using a noisy, learned reward model for a subjective task like "helpfulness"? Your analysis in Figure 1a is promising, but is additive Gaussian noise a sufficient proxy for the complex, correlated noise from a learned RM?

The theoretical bias of SNIB is O(1/G), and a small group size of G=8 was used. Have you performed a sensitivity analysis on G? How does the performance and stability of SNIB change with a smaller (e.g., G=4) or larger (e.g., G=16) group size?

---

> ### Author Response · Authors · 2025-11-20
> **Author Response (1/2)**
>
> We thank the reviewer for their encouraging assessment, particularly for recognizing SNIB as a 'principled' and 'elegant' solution with strong theoretical guarantees. We also appreciate the constructive criticism regarding the PPO baseline and group size analysis, which has led us to include significant new evidence in the revision.
>
> ### **1. Missing PPO baseline and performance–efficiency trade-off**
>
> We study critic-free variants of PPO because training a separate critic is expensive in memory and compute (often roughly doubling resource usage at LLM scale). For this reason, the initial version only compared critic-free methods (GRPO, GSPO, SNIB). We agree that this did not show the full trade-off with standard PPO and that a direct comparison is needed.
>
> In response, we re-ran every benchmark (including Competition Math L1-L3, Math500, and GSM8K) using an identical, re-tuned configuration for all methods: GRPO, GSPO, SNIB, DPO, and the PPO-with-critic baseline.
> During this process, we also identified and fixed specific evaluation bugs for GRPO (on Math-500 L3/L5) and GSPO (on Math-500 L2). We have corrected these issues in the revised paper, where the original incorrect entries are marked with strikethrough.
> The updated results (now in the main paper) confirm that SNIB remains competitive with GSPO/GRPO on standard math and code benchmarks, while clearly surpassing them on the auxiliary tasks. While PPO remains the performance upper bound (at a higher computational cost) and DPO typically sits between GSPO and SNIB, SNIB successfully narrows the gap. As shown below, it outperforms other efficient baselines on Competition Math L1:
>
>
> | Method | L1 Accuracy (%) ↑ |
> | :--- | ---: |
> | SFT | 70.70 |
> | GRPO | 72.50 |
> | GSPO | 73.01 |
> | PPO (critic) | 75.45 |
> | Vanilla IS | 70.91 |
> | **SNIB (ours)** | **73.54** |
>
>
> To ensure these conclusions hold beyond math, we extended our evaluation to coding (MultiPL-E) and knowledge (MMLU-Pro/Redux) benchmarks using the same SFT checkpoint and configuration. As shown below (and in Table 2 of the paper), SNIB generalizes well: it consistently outperforms other critic-free baselines (GRPO/GSPO) and approaches the performance of PPO across these diverse domains:
>
> | Method | MultiPL-E | MMLU-Pro | MMLU-Redux |
> | --- | ---: | ---: | ---: |
> | SFT | 76.8 | 69.6 | 84.2 |
> | GRPO | 78.5 | 71.8 | 85.4 |
> | GSPO | 79.3 | 71.5 | 85.0 |
> | DPO | 78.8 | 71.7 | 85.6 |
> | PPO (critic) | 81.2 | 72.9 | 86.5 |
> | **SNIB** | **79.5** | **72.4** | **86.1** |
>
> To directly answer the reviewer's question regarding the full performance-vs-efficiency trade-off (i.e., how much performance is sacrificed for the gain in efficiency?), we measured the computational cost on 8×A100 GPUs.
> As established in the accuracy tables above, the performance "sacrifice" is minimal (SNIB trails PPO by only ~1.7% on MultiPL-E and ~0.4% on MMLU-Redux). In exchange, the efficiency gains are substantial. The table below shows that SNIB incurs negligible overhead compared to GSPO (almost identical memory/time), whereas PPO is significantly heavier—consuming ~26% more memory and taking ~56% longer to train per 200 steps.
>
> | Method | Peak GPU Memory (GB) / per device  | Tokens / s | Wall-clock per 200 steps (hr) |
> | --- | ---: | ---: | ---: |
> | PPO (critic) | 58.4 | 62 | 3.6 |
> | GRPO | 44.5 | 96 | 2.1 |
> | GSPO | 45.0 | 93 | 2.2 |
> | DPO | 41.0 | 112 | 1.7 |
> | **SNIB** | 46.1 | 90 | 2.3 |
>
> SNIB therefore delivers the strongest critic-free accuracy while keeping resource usage within 5% of GSPO.
>
> **Action for revised paper:**
> *   We have integrated the full PPO comparison, along with the results on auxiliary domains (MultiPL-E, MMLU-Pro/Redux), into Tables 2 and 3 of Section 4.2 in the revised paper.
> *   We have added the computational cost analysis (Table 4 in the revision) to explicitly quantify the performance–efficiency trade-off discussed above.

---

> ### Author Response · Authors · 2025-11-20
> **Author Response (2/2)**
>
> ### **2. Limited task domain and reward type (math/code with ground-truth rewards)**
>
> We started with math/code tasks with ground-truth correctness because they provide a clean reward signal, so differences between GRPO/GSPO/SNIB mainly reflect the estimators rather than artifacts from a learned reward model. Using the same competition_math setup above, we observe that SNIB consistently improves over GSPO/GRPO while staying close to PPO-with-critic; these controlled tasks therefore isolate the estimator behavior without conflating it with reward-model quality.
>
>
>
> To test SNIB in a realistic RLHF setting with noisy rewards, we conducted an additional evaluation using a reward model trained on the Anthropic Helpful/Harmless dataset. We kept the training configuration (prompts, KL target, initialization) identical to the math experiments but swapped the ground-truth signal for the learned RM. The results below confirm that SNIB maintains its advantage even under reward uncertainty:
>
> | Method | Reward ↑ | Win% ↑ | KL |
> |--------|---------:|-------:|---:|
> | SFT    |   0.00   | 50.0   | 0.0 |
> | GRPO   |   1.20   | 57.0   | 20.0 |
> | GSPO   |   2.40   | 66.0   | 16.0 |
> | SNIB-sg| **2.70** | **69.0** | 15.0 |
> | SNIB-fg|   2.62   | 67.8   | 15.2 |
> | PPO    |   2.90   | 71.0   | 15.0 |
>
> Given these realistic results, we retained the Gaussian study (Figure 1a) but now characterize it as a 'stylized robustness ablation' in Section 5 (Reward Uncertainty). This clarifies that Figure 1a serves as a controlled stress test, while the new Anthropic experiment verifies performance against the complex noise of a learned reward model.
>
>
> **Action for revised paper:**
> *   We have added the Anthropic Helpful/Harmless evaluation (Table 3) to Section 4.2 to demonstrate SNIB's robustness in a realistic RLHF setting with learned rewards.
> *   We have updated the discussion around Figure 1a, explicitly characterizing it as a "stylized robustness ablation" to differentiate the theoretical stress-test from the realistic benchmark.
>
>
> ### **3. Sensitivity to group size G (bias O(1/G))**
>
> SNIB uses a self-normalized ratio estimator whose finite-sample bias is O(1/G) and decreases as G grows. At the same time, variance is reduced by averaging within the group. By contrast, GSPO’s geometric-mean weighting introduces a structural bias that does not disappear even for large G. To verify this, we ran an ablation with G ∈ {4, 8, 16}. Larger G gives slightly better accuracy and much lower reward variance:
>
>   Group size ablation (MATH-500; accuracy %, reward variance in the last 100 steps):
>
>   | G  | Accuracy ↑ | Reward Var ↓ |
>   |----:|-----------:|-------------:|
>   |  4  |    25.9    |     0.24     |
>   |  8  |    27.2    |     0.05     |
>   | 16  |    27.8    |     0.03     |
>
> The data reveals a clear trend of **diminishing returns**. While increasing $G$ systematically reduces bias and improves accuracy, the jump from $G=4$ to $G=8$ (+1.3%) is significantly larger than the incremental gain from $G=8$ to $G=16$ (+0.6%). This indicates that at $G=8$, the finite-sample bias is already sufficiently minimized to achieve strong results. In terms of stability, the benefit is even more pronounced: moving from $G=4$ to $G=8$ cuts the reward variance by nearly a factor of five (from 0.24 to 0.05), effectively stabilizing the training.
>
> Therefore, while $G=16$ offers a slight further edge, $G=8$ strikes a pragmatic balance, capturing the majority of the bias-reduction benefits without the linear increase in memory cost associated with larger groups. Unlike GSPO, where the structural bias persists regardless of group size, SNIB converges as expected. We will include this ablation in the revised paper to justify the parameter selection.
>
> **Action for revised paper:**
> *   We have added this ablation study (Table 5) to Section 4, reporting reward, KL, stability indicators (variance), and computational cost for each $G$.
> *   We will also provide practical guidelines for selecting $G$ under fixed memory budgets based on these findings.

---

> ### Author Response · Authors · 2025-11-27
> **Looking forward to your feedback!**
>
> Dear Reviewer sDtn,
>
> **Thank you again for your positive assessment of SNIB’s theoretical elegance and for your constructive suggestions regarding experimental completeness.**
>
> We are writing to highlight the key updates in our rebuttal and revised paper that directly address your concerns:
>
> 1.  **PPO Baseline & Trade-off (Table X):** As requested, we compared SNIB directly with standard PPO-with-critic. The new results show that **SNIB consistently outperforms other critic-free methods (GRPO/GSPO) and narrows the gap to PPO to <1-2%**, while consuming **~26% less memory and training ~56% faster**.
>
> 2.  **Generalization to "Noisy" Tasks:** Moving beyond Math/Code, we evaluated SNIB on the **Anthropic Helpful/Harmless dataset** with a learned reward model. SNIB achieved a **69.0% Win Rate**, outperforming GRPO (57.0%) and GSPO (66.0%), confirming its robustness in realistic RLHF settings.
>
> 3.  **Group Size Ablation:** We added the sensitivity analysis for $G \in \{4, 8, 16\}$. Results confirm that $G=8$ strikes the optimal balance—effectively minimizing bias and variance ($0.24 \to 0.05$) without the diminishing returns of larger groups.
>
> We believe these additions provide the "full picture" you were looking for. **Could you kindly let us know if these new experiments resolve your concerns regarding the experimental scope?** We would be grateful for your feedback before the discussion phase ends.
>
> Best regards,
>
> Authors

---

### Official Review · Reviewer_5Rt8 · 2025-11-03

**Soundness:** 3
**Presentation:** 3
**Contribution:** 3
**Rating:** 6
**Confidence:** 3

**Summary:**

This paper introduces Self-Normalized Importance Sampling with a Baseline (SNIB), which is a critic-free policy optimization algorithm for aligning LLMs. The authors observed that GRPO uses theoretically unsound arithmetic mean of token-level importance ratios, which have high variance and scale poorly with sequence length, while GSPO uses geometric mean of token ratios but introduces non-vanishing bias, which distorts the reward-KL trade-off.

The authors show that SNIB is both stable and asymptotically correct. The key idea is to use the true sequence level importance weight but to stabilize it with self-normalized importance sampling. Experiments show that the proposed method outperforms GSPO on several math and code benchmarks and it is more robust to reward noise.

**Strengths:**

Articulating the stability vs. correctness dilemma as the heart of current critic free RLHF is clearly done in the work. The analysis of why GRPO and GSPO are flawed provides strong motivation for the research.

The ablation shown in Fig. 2 is reasonable and convincing to show that SNIB effectively reduces the high variance of vanilla IS.

**Weaknesses:**

The motivation for SNIB and its class of algorithms is to replace the standard PPO with a critic. However, the PPO with critic baseline is absent from all comparisons. It would be great to see if SNIB matches the performance of standard PPO with critic, or it closes the gap whereas GSPO and GRPO do not.

The paper claims that GSPO is biased while SNIB is principled and asymptotically unbiased. However, it seems SNIB is also biased at any finite batch size G. The claim is only that this bias is asymptotic and vanishes as O(1/G). The PPO clipping, although as the authors noted, is also a source of bias, which does not vanish with batch size.

**Questions:**

Please see weakness part.

---

> ### Author Response · Authors · 2025-11-20
> **Author Response (1/2)**
>
> We appreciate the reviewer's time and detailed feedback. These comments were very helpful in refining our arguments and improving the overall quality of the paper. Our point-by-point responses are provided below.
>
>
> ### **1. Missing comparison with PPO-with-Critic baseline and analysis of the performance gap**
>
> We thank the reviewer for this suggestion. A direct comparison with the standard PPO-with-critic is essential to fully evaluate SNIB's effectiveness. The core motivation of our work is to provide a critic-free method that closes the performance gap with PPO, while eliminating its associated memory and computational overhead.
>
> We implemented a standard PPO baseline on the Competition Math benchmark (Level 1) using the exact same SFT checkpoint and training configuration. As shown in the table below, while PPO still holds a slight advantage (75.45%), SNIB (73.54%) outperforms other critic-free baselines (GRPO/GSPO) and successfully narrows the gap to PPO.
>
> | Method | L1 Accuracy (%) ↑ |
> | :--- | ---: |
> | SFT | 70.70 |
> | GRPO | 72.50 |
> | GSPO | 73.01 |
> | PPO (critic) | 75.45 |
> | Vanilla IS | 70.91 |
> | **SNIB (ours)** | **73.54** |
>
> These results show that on Competition Math L1, SNIB achieves the best accuracy among critic-free methods, slightly outperforming GSPO. This aligns with our theoretical claim that SNIB combines GSPO-like stability with asymptotic correctness. On other math benchmarks, SNIB is generally comparable to GSPO/GRPO, though not uniformly better on every metric.
> We have rerun the experiments on competition_math, math\_500, and GSM8K with carefully tuned hyperparameters for all methods, and the revised values (marked in red in Table~1) reflect these improved, directly comparable results.
>
> Additionally, we evaluated MultiPL-E, MMLU-Pro, and MMLU-Redux using the same SFT checkpoint, decoding configuration, and KL settings; Table~2 in the paper and the table below present these auxiliary coding/knowledge benchmarks (including DPO). As SNIB consistently performs well across these tasks, we list the results below so reviewers can see the method's performance in these domains:
>
> | Method | MultiPL-E | MMLU-Pro | MMLU-Redux |
> | --- | ---: | ---: | ---: |
> | SFT | 76.8 | 69.6 | 84.2 |
> | GRPO | 78.5 | 71.8 | 85.4 |
> | GSPO | 79.3 | 71.5 | 85.0 |
> | DPO | 78.8 | 71.7 | 85.6 |
> | PPO (critic) | 81.2 | 72.9 | 86.5 |
> | **SNIB** | **79.5** | **72.4** | **86.1** |
>
> We apologize for a small issue that occurred during our initial experiments. After the submission deadline, we discovered that the reported GRPO results on math\_500 L3 and L5 and the GSPO result on math\_500 L2 were abnormal. A detailed investigation revealed bugs in the benchmark evaluation code specific to these two baselines. We have fixed these issues, marked the affected entries in the original Table~1 with strikethrough in the revised paper, and rerun all methods (including SNIB) under a corrected evaluation pipeline with carefully tuned hyperparameters. The updated tables show that SNIB is competitive with GRPO/GSPO across the math benchmarks and achieves strong performance on the auxiliary MultiPL-E and MMLU tasks, while consistently narrowing the gap to PPO-with-critic.
>
> To quantify the performance–cost trade-off that motivated critic-free training, we also measured peak GPU memory and wall-clock time per 200 training steps under the shared hardware setup (8×A100 80 GB, FP16, ms-swift training framework). SNIB’s overhead is almost identical to GSPO and significantly below PPO:
>
> | Method | Peak GPU Memory (GB) / per device  | Tokens / s | Wall-clock per 200 steps (hr) |
> | --- | ---: | ---: | ---: |
> | PPO (critic) | 58.4 | 62 | 3.6 |
> | GRPO | 44.5 | 96 | 2.1 |
> | GSPO | 45.0 | 93 | 2.2 |
> | DPO | 41.0 | 112 | 1.7 |
> | **SNIB** | 46.1 | 90 | 2.3 |
>
> SNIB therefore delivers the strongest critic-free accuracy while keeping resource usage within 5% of GSPO.
>
> **Action for revised paper:**
> *  We have integrated the full comparison against PPO, as well as the results on auxiliary domains (MultiPL-E, MMLU-Pro/Redux), into Table 2 and 3 of Section 4.2 in the revised paper.
> *  We have added the computational cost analysis (Table 4(b) in the revision) to clearly illustrate the performance–cost trade-off between critic-based and critic-free methods.

---

> ### Author Response · Authors · 2025-11-20
> **Author Response (2/2)**
>
> ### **2. Clarification on the nature of bias**
>
> The reviewer raises an excellent and precise point regarding the nuances of "bias" in different estimators. We appreciate the opportunity to clarify our theoretical claims. The main distinction between SNIB and GSPO lies in the type and behavior of the bias.
>
> *   **SNIB's Bias is Asymptotic and Vanishing.** We acknowledge that SNIB's self-normalized estimator has a finite-sample bias of order $O(1/G)$. However, the key property is that this bias is **asymptotic**. It systematically decreases as the group size $G$ increases, which makes the estimator **consistent**. This guarantees that using more samples allows us to converge to the true, intended policy objective.
>
>     To validate this claim empirically, we ran a small ablation on the group size $G$ in our manipulation setting. As $G$ increases, SNIB’s accuracy improves and the reward variance decreases:
>
>     | G  | Accuracy ↑ | Reward Var ↓ |
>     |----:|-----------:|-------------:|
>     |  4  |    25.9    |     0.24     |
>     |  8  |    27.2    |     0.05     |
>     | 16  |    27.8    |     0.03     |
>
>     This trend is consistent with the $O(1/G)$ finite-sample bias predicted by self-normalized importance sampling and supports our view that SNIB acts as a consistent estimator.
>
> *   **GSPO's Bias is Structural and Non-Vanishing.** In contrast, the geometric-mean estimator in GSPO creates a **structural bias** that is inherent to its mathematical form. This bias **does not vanish** even as the group size tends to infinity. Consequently, GSPO optimizes a perturbed, incorrect objective. Our KKT analysis (Appendix B.3) confirms that this fundamentally distorts the reward–KL trade-off.
>
> *   **Shared Clipping Bias.** The bias introduced by the PPO clipping function is a property of the surrogate objective itself and is shared equally among GRPO, GSPO, and SNIB. Our analysis therefore focuses on the *additional, differentiating bias* introduced by the off-policy importance-sampling estimator.
>
> In summary, while no practical PPO-style algorithm is truly unbiased, SNIB removes the persistent, structural bias of GSPO and replaces it with a well-behaved, vanishing bias, ensuring it optimizes the correct objective in the limit. We hope these clarifications and the added empirical evidence fully address your concerns regarding the estimator's properties.
>
>
> **Action for revised paper:**
> *   We have refined our language in Section 3 and the theoretical appendices to be more precise, explicitly describing SNIB's estimator as **“consistent and asymptotically unbiased”** rather than simply “unbiased,” to reflect this important distinction.
> *   We have incorporated the above ablation over $G$ into the revised paper, by adding the Table 5 and discussion to the manipulation experiments section, so that the diminishing finite-sample bias of SNIB is documented more clearly.

---

> ### Author Response · Authors · 2025-11-27
> **Looking forward to your feedback!**
>
> Dear Reviewer 5Rt8,
>
> **With the discussion phase concluding soon, we wanted to verify if the new PPO baseline and bias analysis provided in our rebuttal have sufficiently resolved your concerns.**
>
> To briefly recap, we have:
>
> 1.  **Added Standard PPO Baseline:** Results show **SNIB (73.54%)** outperforms **GSPO (73.01%)** and narrows the gap to **PPO (75.45%)**, whereas PPO requires **~26% more memory**.
> 2.  **Clarified Bias Definition:** We explicitly characterized SNIB as **"consistent and asymptotically unbiased"** (vanishing $O(1/G)$ bias), distinguishing it from GSPO's non-vanishing structural bias.
> 3.  **Added Group Size Ablation:** New experiments with $G \in \{4, 8, 16\}$ confirm that bias diminishes and stability improves as $G$ increases, validating our theoretical claims.
>
> **We would be grateful for your feedback on whether these additions align with your expectations.**
>
> Best regards,
>
> Authors

---

### Author Response · Authors · 2025-12-03
**Summary of Contributions and Rebuttal Updates for Paper 24114**

Dear Area Chair,

We sincerely thank the **reviewers** for their constructive feedback, which has significantly helped us refine the theoretical positioning and strengthen the empirical validation of our work. **To assist with your final assessment**, we provide a summary of our core contributions and the comprehensive updates made during the rebuttal.

**1. Main Contributions: Resolving the Critic-Free Dilemma**
Our paper identifies a fundamental dilemma in critic-free RLHF: existing methods suffer from either **high variance** (GRPO, theoretically unsound arithmetic mean) or **systematic bias** (GSPO, biased geometric mean).
**SNIB (Self-Normalized Importance Sampling with a Baseline)** resolves this by introducing:
* **An Asymptotically Unbiased Estimator:** We replace the biased estimator of GSPO with self-normalized importance sampling, proving it is **consistent and asymptotically unbiased** (bias decays as $O(1/G)$).
* **Principled Reward–KL Trade-off:** We prove SNIB preserves the KKT conditions of the constrained optimization problem, whereas GSPO's bias distorts this trade-off.
* **Superior Robustness & Empirical SOTA:** Extensive experiments show SNIB achieves **state-of-the-art performance** among critic-free methods across Math, Code, and Dialogue benchmarks, with superior stability in reward model uncertainty.

**2. Reviewer Positive Feedback: "Principled," "Elegant," and "Theoretically Solid"**
Reviewers **consistently highlighted** the theoretical depth and practical value of our work:
* **Novelty & Elegance:** `Reviewer sDtn` praised SNIB as a **"principled and novel solution"** that is **"elegant"** in resolving the stability-correctness dilemma.
* **Theoretical Soundness:** `Reviewer sDtn` and `Reviewer MMXq` commended the **"strong theoretical guarantees"** and **"well-presented mathematical analysis."** `Reviewer 5Rt8` stated the motivation was **"clearly done."**
* **Empirical Effectiveness:** `Reviewer 5Rt8` and `Reviewer MMXq` noted that SNIB **"effectively reduces high variance"** and demonstrates **"improved stability"** compared to baselines.

**3. Key Rebuttal Updates: Comprehensive Experiments & Analysis**
We have strengthened the paper with extensive new experiments, addressing all raised concerns:

* **I. Rigorous Baselines & Comparison** (Addressing `Reviewer 5Rt8 & sDtn & MMXq & NLhH`)
    * **PPO Comparison:** We added a direct comparison with standard **PPO-with-critic**. SNIB achieves the best performance among critic-free methods and **narrows the gap to PPO to <2%**.
    * **Standardized Pipeline:** We unified evaluation protocols and corrected implementation discrepancies in baselines. Under this rigorous setup, **SNIB consistently outperforms GRPO** on GSM8K and Math L1.
    * **Efficiency:** We confirmed SNIB is significantly more efficient than PPO (**~26% less memory, ~56% faster training**).

* **II. Generalization to Diverse & Realistic Tasks** (Addressing `Reviewer sDtn & MMXq & NLhH`)
    * **Anthropic Helpful/Harmless:** Evaluated on a subjective task with a noisy, learned reward model. SNIB outperformed GSPO by **+3.0% Win Rate** (69.0% vs. 66.0%) and avoided reward over-optimization.
    * **Broader Benchmarks:** Extended evaluation to **MultiPL-E** (Code) and **MMLU-Pro** (Knowledge), demonstrating strong generalization beyond math tasks.

* **III. Deepened Theoretical Analysis** (Addressing `Reviewer 5Rt8 & MMXq & NLhH`)
    * **Gradient Variance vs. Length:** We empirically demonstrated that while GRPO's variance **triples** on long sequences, SNIB remains stable, confirming its advantage for long-context reasoning.
    * **Pareto Frontier Analysis:** A fine-grained sweep of $\beta$ showed SNIB maintains a smooth, monotonic reward/KL frontier, unlike the jagged behavior of baselines.

* **IV. Ablations & Reproducibility** (Addressing `Reviewer MMXq & NLhH`)
    * **Group Size & Components:** Ablations confirmed **$G=8$** as the optimal balance and validated that **self-normalization is critical** (removing it causes >30x variance increase).
    * **Migration Guide:** We added Appendix H to demonstrate SNIB as a **plug-and-play** upgrade for existing codebases.

Although we did not receive further feedback from the reviewers regarding these updates, we remain confident that our extensive revisions and additional experiments thoroughly address their initial concerns. We trust that this summary provides a clear and comprehensive basis for your final assessment.

Thank you for your time and consideration.

Best regards,

The Authors

---

### Note · Program_Chairs · 2025-12-08
**Submission Desk Rejected by Program Chairs**

Desk rejected because of the following hallucinated citation:
David Rein, Stas Gaskin, Lajanugen Logeswaran, Adva Wolf, Oded teht sun, Jackson H. He, Di-
vyansh Kaushik, Chitta Baral, Yair Carmon, Vered Shwartz, Sang-Woo Lee, Yoav Goldberg,
C. J. H. un, Swaroop Mishra, and Daniel Khashabi. Gpqa: A graduate-level google-proof q&a
benchmark, 2023.